# Function of the SNARE Ykt6 on autophagosomes requires the Dsl1 complex and the Atg1 kinase complex

Jieqiong Gao[1], Rainer Kurre[2,3], Jaqueline Rose[1], Stefan Walter[2], Florian Fröhlich[4] (iD), Jacob Piehler[2,3,5], Fulvio Reggiori[6] (iD) & Christian Ungermann[1,2,*] (iD)

## Abstract

The mechanism and regulation of fusion between autophagosomes and lysosomes/vacuoles are still only partially understood in both yeast and mammals. In yeast, this fusion step requires SNARE proteins, the homotypic vacuole fusion and protein sorting (HOPS) tethering complex, the RAB7 GTPase Ypt7, and its guanine nucleotide exchange factor (GEF) Mon1-Ccz1. We and others recently identified Ykt6 as the autophagosomal SNARE protein. However, it has not been resolved when and how lipid-anchored Ykt6 is recruited onto autophagosomes. Here, we show that Ykt6 is recruited at an early stage of the formation of these carriers through a mechanism that depends on endoplasmic reticulum (ER)-resident Dsl1 complex and COPII-coated vesicles. Importantly, Ykt6 activity on autophagosomes is regulated by the Atg1 kinase complex, which inhibits Ykt6 through direct phosphorylation. Thus, our findings indicate that the Ykt6 pool on autophagosomal membranes is kept inactive by Atg1 phosphorylation, and once an autophagosome is ready to fuse with vacuole, Ykt6 dephosphorylation allows its engagement in the fusion event.

**Keywords** autophagy; COPII vesicles; Dsl1 complex; SNARE; Ykt6
**Subject Categories** Autophagy & Cell Death; Membrane & Trafficking

## Introduction

Macroautophagy (called hereafter autophagy) is a highly conserved degradation pathway that contributes to maintain cellular homeostasis in eukaryotic cells by recycling intracellular components and using the resulting degradation products such as amino acids for the synthesis of new macromolecules or as a source of energy. During autophagy, proteins, protein complexes, and organelles are sequestered within double-membrane vesicles, known as autophagosomes (Nakatogawa, 2020). Autophagosome formation is a complex process that is orchestrated by the autophagy-related (Atg) proteins and begins with the nucleation of a cisterna known as the phagophore or isolation membrane. Expansion and closure of the phagophore lead to the formation of an autophagosome. Once complete, autophagosomes fuse with lysosomes in mammalian cells and with vacuoles in yeast and plants, and the autophagosomal cargo material is degraded in the interior of these hydrolytic compartments (Reggiori & Ungermann, 2017; Zhao & Zhang, 2018).

The endoplasmic reticulum (ER) has been proposed to be the main origin for the lipids constituting the autophagosomal membranes (Hayashi-Nishino et al, 2009; Ylä-Anttila et al, 2009). Recent studies also suggested additional membrane sources such as the ER–Golgi intermediate compartments (Ge & Schekman, 2014), the Golgi (van der Vaart et al, 2010), mitochondria (Hailey et al, 2010), the plasma membrane (Ravikumar et al, 2010), the recycling endosomes (Longatti et al, 2012; Puri et al, 2013), and the transmembrane Atg9-containing vesicles (Mari et al, 2011; Yamamoto et al, 2012), but those appear to be more crucial for the phagophore formation. In yeast, autophagosome biogenesis occurs at one perivacuolar location known as the phagophore assembly site or preautophagosomal structure (PAS), where the Atg1 protein kinase and its binding partner Atg13 act as a central regulator and/or scaffold (Matsuura et al, 1997; Noda & Fujioka, 2015; Fujioka et al, 2020). In yeast, the generation and early expansion of the phagophore appear to require COPII-coated vesicles and functional ER exit sites, where COPII-coated vesicles originate from (Jensen & Schekman, 2011; Graef et al, 2013; Suzuki et al, 2013; Wang et al, 2014; Shima et al, 2019). Additionally, earlier studies suggested that Atg9, the only transmembrane protein among all core Atg proteins, cycles between a compartment named as the Atg9 reservoirs and the PAS, and it is also key for the phagophore formation (Mari et al, 2011; Reggiori &

1 Department of Biology/Chemistry, Biochemistry Section, University of Osnabrück, Osnabrück, Germany
2 Center of Cellular Nanoanalytics Osnabrück (CellNanOs), University of Osnabrück, Osnabrück, Germany
3 Center of Cellular Nanoanalytics, Integrated Bioimaging Facility, University of Osnabrück, Osnabrück, Germany
4 Department of Biology/Chemistry, Molecular Membrane Biology Group, University of Osnabrück, Osnabrück, Germany
5 Department of Biology/Chemistry, Biophysics Section, University of Osnabrück, Osnabrück, Germany
6 Department of Biomedical Sciences of Cells and Systems, University of Groningen, University Medical Center Groningen, Groningen, Netherlands
*Corresponding author. Tel: +49 541 969 2752; E-mail: cu@uos.de

Tooze, 2012; Yamamoto *et al*, 2012; Rao *et al*, 2016; Matscheko *et al*, 2019). Recently, Atg2, an Atg9 interactor, has been identified as a possible lipid slide that directly connects the ER and the phagophore, and may thus provide lipids for the nascent autophagosome (Gómez-Sánchez *et al*, 2018; Kotani *et al*, 2018; Osawa *et al*, 2019; Valverde *et al*, 2019). The individual contribution of each of these membrane sources, however, has not been clarified yet.

Fusion of autophagosomes with the lysosome requires the RAB7 GTPase, its GEF Mon1-Ccz1, SNAREs, and the membrane tethering HOPS complex (Barr, 2013; Kümmel & Ungermann, 2014; Nakamura & Yoshimori, 2017; Reggiori & Ungermann, 2017; Bas *et al*, 2018; Gao *et al*, 2018a). The RAB7 GTPase is crucial for autophagosome–lysosome fusion (Hegedüs *et al*, 2016). We recently showed that yeast Mon1-Ccz1 associates with autophagosomes by binding Atg8 and specifically recruits the Rab7-like Ypt7 to this location (Gao *et al*, 2018a). Ypt7-GTP then interacts with the HOPS tethering factor to promote SNARE-mediated fusion (Bas *et al*, 2018; Gao *et al*, 2018a). SNAREs are membrane-resident fusion proteins, which are present on opposing membranes and assemble into specific coiled-coil SNARE complexes during fusion (Jahn & Scheller, 2006). The four SNARE domains interact via several hydrophobic layers and one hydrophilic layer, in which three SNARE domains contribute a glutamine (Q-SNARE) and one an arginine (R-SNARE) (Weimbs *et al*, 1997; Fasshauer *et al*, 1998).

Recently, we and others identified Ykt6 as the autophagosomal R-SNARE, which functions alone in yeast, while it has redundant function with SYNTAXIN17 in metazoan cells (Bas *et al*, 2018; Gao *et al*, 2018b; Kriegenburg *et al*, 2019; Mizushima *et al*, 2019). Ykt6 contains an N-terminal longin domain, which can interact with the central SNARE domain (Tochio *et al*, 2001; Pylypenko *et al*, 2008). Unlike most of the other SNAREs, Ykt6 does not possess a transmembrane segment, and it is posttranslationally modified by dual prenylation, farnesylation, and—as recently reported—geranylgeranylation, to stay associated with membranes (McNew *et al*, 1997; Dietrich *et al*, 2004, 2005; Fukasawa *et al*, 2004; Shirakawa *et al*, 2020). Ykt6 is also involved in the retrograde trafficking from the cis-Golgi to the ER, as well as in different membrane trafficking events at endosomes and vacuoles (McNew *et al*, 1997; Kweon *et al*, 2003; Bas *et al*, 2018; Gao *et al*, 2018b; Matsui *et al*, 2018). However, it remains unknown how Ykt6 is specifically targeted to autophagosomes and how its function in fusion is timely regulated.

We now show that COPII-coated vesicles and the Dsl1 complex are key for the specific recruitment of Ykt6 to the PAS, and that Ykt6 is negatively regulated by the Atg1 kinase complex to guarantee its functioning, when an autophagosome is complete and ready to fuse.

# Results

## Ykt6 localizes to the ER, vacuoles, and endosomes under autophagy-inducing conditions

Ykt6 is required on autophagosomes for fusion with the vacuole, but it remains unknown how this SNARE is recruited onto these vesicles (Bas *et al*, 2018; Gao *et al*, 2018b; Kriegenburg *et al*, 2019; Mizushima *et al*, 2019). To understand how Ykt6 is targeted to autophagosomes, we first determined the distribution of Ykt6

under autophagy-inducing conditions. Our previous studies showed that Ykt6 appears to be principally cytosolic in nutrient-rich conditions (Meiringer *et al*, 2008), but localizes to several distinct puncta upon nitrogen starvation, some of which being autophagosomes (Gao *et al*, 2018b). In yeast, autophagosomes are generated at the PAS, which is adjacent to both the ER and the vacuole (Suzuki *et al*, 2001, 2013; Kim *et al*, 2002; Graef *et al*, 2013). To examine Ykt6 subcellular redistribution upon autophagy induction, we stimulated autophagy by nitrogen starvation and analyzed colocalization of overproduced GFP-tagged Ykt6 with BFP-tagged Ape1, a specific autophagosomal cargo protein that can also serve as a PAS marker (Kim *et al*, 1997), and tdTomato-tagged Sec63, an ER-resident protein (Rothblatt *et al*, 1989). Overproduction of Ykt6 facilitates the analysis of its distribution, especially on autophagosomes and thus also on autophagosomal intermediates (Gao *et al*, 2018b).

Over time, we observed an increase in GFP-Ykt6 puncta that colocalized with Ape1 proximal to the ER (Fig 1A and B). Ykt6 has been found at other organelles under growing conditions (Meiringer *et al*, 2008). We thus examined the colocalization of GFP-tagged Ykt6 and tdTomato-tagged Snx41 (endosome), tdTomato-tagged Sec63 (ER), tdTomato-tagged Vac8 (vacuole), or tdTomato-tagged Mnn9 (Golgi) (Jungmann & Munro, 1998; Wang *et al*, 1998; Hettema *et al*, 2003). We observed puncta positive for Ykt6 and the marker proteins of the ER, vacuoles, and endosomes, whereas colocalization with the Golgi marker protein was less prominent (Fig 1C). These data suggest that in addition to be detected at the Ape1-positive structures, Ykt6 is also present on other organelles under autophagy-inducing conditions, and these organelles might provide membranes to the PAS.

## Ykt6 is recruited to the PAS prior to autophagosome completion

Since Ykt6 localizes to different organelles, also under starvation conditions, we were wondering how Yk6 is targeted to autophagosomal membranes. We reasoned that like for the Mon1-Ccz1 GEF (Gao *et al*, 2018a), Ykt6 recruitment to the PAS should depend on one or more Atg proteins. Thus, we analyzed the colocalization of GFP-Ykt6 relative to mCherry-Ape1 in the absence of *ATG* genes that block different steps of the autophagosome biogenesis. We opted for Ape1 as the PAS marker protein instead of Atg8, because the recruitment of this latter protein to the PAS is impaired in several *atg* mutants (Suzuki *et al*, 2007). GFP-Ykt6 was not detected at the PAS in knockout cells such as *atg1Δ*, *atg13Δ*, and *atg17Δ* (Fig 2A and B), which block the initiation of autophagy due to the lack of a functional Atg1 kinase complex (Kabeya, 2005). In contrast, GFP-Ykt6 recruitment to this location was partially or not affected in the rest of the examined mutants (Fig 2A and B). These data indicate that Ykt6 is targeted to autophagosomal membranes before autophagosome completion and that this recruitment depends on the Atg1 kinase complex. We noticed that Ykt6 was not present on the PAS in *atg17Δ* cells, and it is known that cytoplasm to vacuole (Cvt) pathway does not require Atg17 (Cheong *et al*, 2005). We thus asked whether Ykt6 is necessary for the Cvt pathway, and thus analyzed for Ape1 processing in wild type, *atg13Δ*, and *ykt6-11* cells (Gao *et al*, 2018b) at restrictive (37°C) temperature (Fig EV1). Whereas *atg13Δ* cells were completely defective, we observed a partial Ape1-processing defect in *ykt6* mutants at the

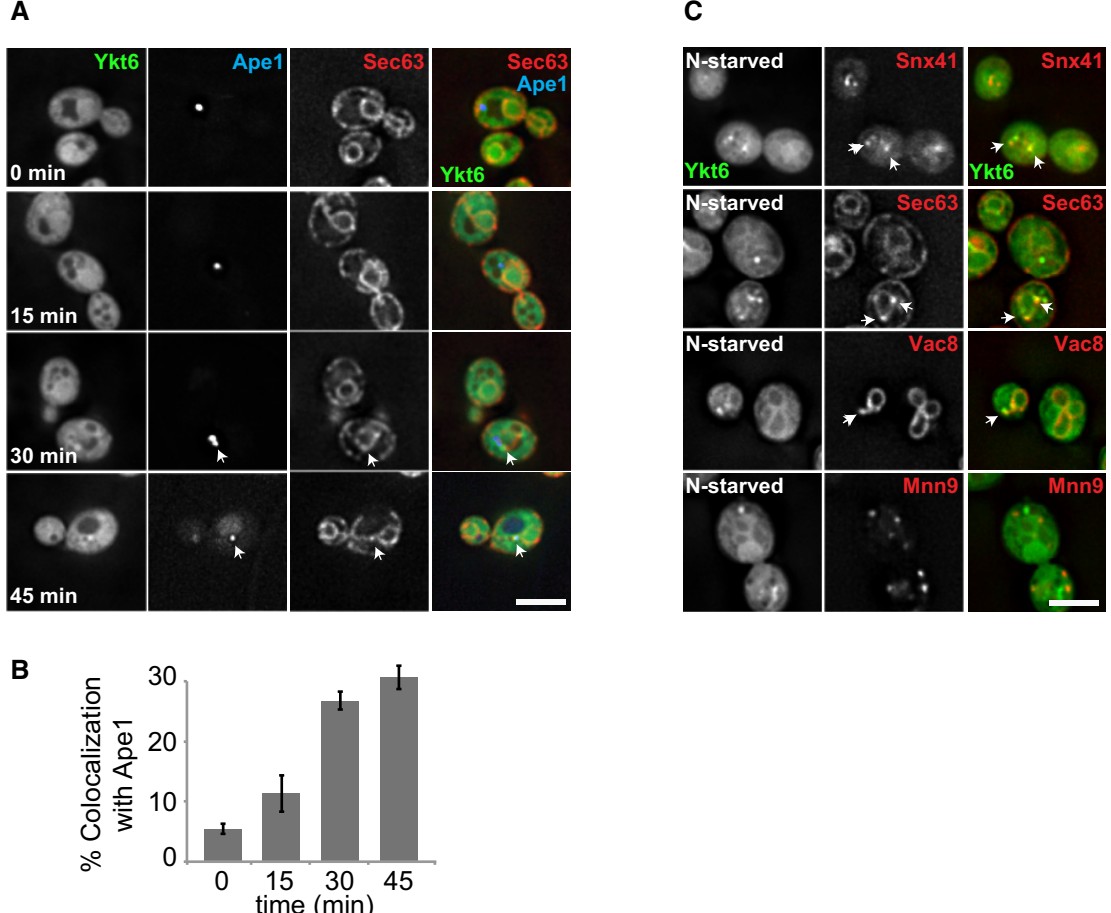

**Figure 1. Ykt6 is located to different organelles during starvation.**

A   Localization of Ape1 relative to Ykt6 and Sec63 during nitrogen starvation. Cells expressing GFP-tagged Ykt6 under the control of the *GAL1* promoter and tdTomato-tagged Sec63 were transformed with a centromeric plasmid expressing BFP-Ape1. Cells were grown in SGC-LEU, shifted to SG-N for the indicated time, and analyzed by fluorescence microscopy. Images are shown as single planes. Scale bar, 5 μm. Arrows indicate Ykt6 dots colocalizing with Ape1.

B   Percentage of Ykt6 puncta simultaneously colocalizing with both Ape1 and Sec63. The data were quantified from (A). Error bars represent standard deviation of three independent experiments. Cells (*n* ≥ 250) and Ape1 dots (*n* ≥ 50) were quantified.

C   Localization of Ykt6 relative to different organelle marker proteins during nitrogen starvation. Cells expressing GFP-tagged Ykt6 under the control of the *GAL1* promoter and tdTomato-tagged Snx41 (endosomes), Sec63 (ER), Vac8 (vacuole), or Mnn9 (Golgi) were grown in SGC-LEU, then shifted to SG-N for 2 h, and analyzed by fluorescence microscopy. Single imaging planes are shown. Scale bar, 5 μm. Arrows indicate Ykt6 dots that colocalize with marker proteins.

restrictive temperature. It is thus possible that other SNAREs can substitute for Ykt6 in the Cvt pathway.

To confirm that GFP-Ykt6 associates with the autophagosomal membrane during the first stages of autophagosome biogenesis, we colocalized Ykt6 with Atg9, the only transmembrane Atg protein that is part of the core Atg machinery (Noda *et al*, 2000). Indeed, Ykt6 puncta were positive for Atg9 only after starvation, also suggesting that Atg9 and Ykt6 probably arrive through different transport routes (Fig 2C).

To confirm that Ykt6 is indeed present on nascent autophagosomes and to also determine Ykt6 distribution on phagophores, we examined the localization of GFP-Ykt6 upon BFP-Ape1 overexpression. This leads to the formation of a giant Ape1 oligomer that is too large to be enclosed by a phagophore, but provides the possibility to better resolve visually the structure of this intermediate (Suzuki *et al*, 2013). Under these conditions, GFP-Ykt6 was found along the

large phagophore marked by mCherry-Atg8 and adjacent to giant BFP-Ape1 (Fig 2D), confirming that Ykt6 is recruited to the autophagosomal intermediates at an early stage and that it does not have a peculiar distribution on phagophores.

Lattice light-sheet microscopy (LLSM) is the method of choice for time-lapse volumetric colocalization with uncompromised spatial and temporal resolution at very low fluorescent signals (Chen *et al*, 2014; Aguet *et al*, 2016; Adell *et al*, 2017). To precisely determine when Ykt6 is recruited to the PAS, we monitored the recruitment and distribution of GFP-Ykt6 on phagophores adjacent to the giant BFP-Ape1 in a time course setup by LLSM. We found that Ykt6 colocalized with mCherry-Atg8 as soon as Atg8 appeared as a puncta (Fig 2F, event 1 from 90 to 210 s, event 2 from 270 to 600 s), and followed its growth around the giant Ape1 structure over time by 3D view (Fig 2E and F (event 2 from 270 to 600 s), Movies EV1 and EV2). We observed that Ykt6 was transported to the PAS and this

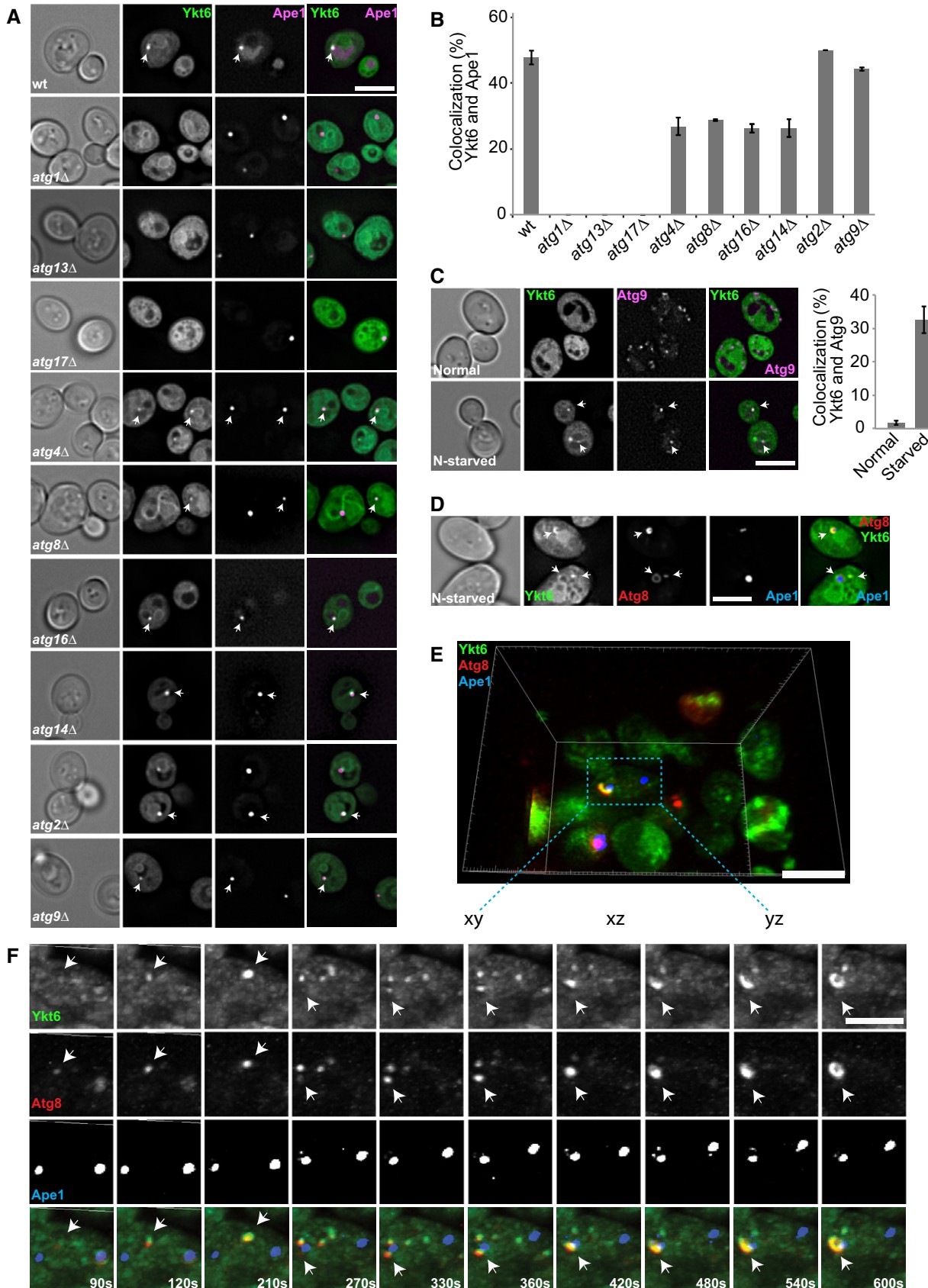

**Figure 2.**

**Figure 2. Ykt6 is recruited to the PAS at an early stage.**

A, B   Localization of Ykt6 in multiple *atg* mutants. (A) Wild-type or the respective mutant cells expressing mCherry-tagged Ape1 and GFP-tagged Ykt6 were grown in SGC to allow for Ykt6 expression from the *GAL1* promoter and analyzed as in Fig 1A. Scale bar, 5 μm. (B) Quantification of colocalization of Ykt6 relative to Ape1 as a PAS marker. The data were quantified from (A). Error bars represent standard deviation of three independent experiments. Cells ($n \geq 250$) and Ape1 dots ($n \geq 70$) were quantified.

C   Localization of Atg9 relative to Ykt6. Cells expressing GFP-tagged Ykt6 and mScarlet-tagged Atg9 were grown and analyzed as in Fig 1. Scale bar, 5 μm. The left is the quantification of colocalization of Ykt6 relative to Atg9. Error bars represent standard deviation of three independent experiments. Cells ($n \geq 200$) and Atg9 dots ($n \geq 80$) were quantified.

D–F   Ykt6 localizes to the isolation membrane during starvation. (D) Cells expressing mCherry-tagged Atg8 and GFP-Ykt6 carrying the Ape1 encoding plasmid pRS315-*CUP1pr-BFP-APE1* were grown in SGC medium containing CuSO$_4$ and starved for 1 h, before to be analyzed by microscopy. Scale bar, 5 μm. (E) Volume rendering of LLSM image from a single time point after 3D deconvolution from 3D view. Cells expressing mCherry-tagged Atg8 and GFP-Ykt6 carrying plasmid pRS315-*CUP1pr-BFP-APE1*, cells were grown as Fig 2D, and images were analyzed by Imaris. Scale bar, 5 μm. (F) The only partial time points (90–600 s) of images from (E) were assembled and analyzed by Imaris and ImageJ. Scale bar, 5 μm. 500–1,000 cells were analyzed in each independent experiment. The 3D stacks were cropped by Imaris, and different channels were split by ImageJ.

Data information: Arrows in the images indicate dots of Ykt6 colocalizing with Ape1 (A), Atg9 (C), and Atg8 (D, E).

event continues over time, as the fluorescence signal increases along the growing giant Ape1 structure. This observation also reiterates the notion that Ykt6 is present at the PAS before autophagosome completion.

## The Dsl1 complex is required for Ykt6 targeting to the PAS

The Dsl1 complex (Dsl1, Dsl3, and Tip20) plays an important role in the retrograde vesicular transport from the Golgi to the ER. It supports the fusion between COPI-coated vesicles and the ER by binding the SNAREs Ufe1, Use1, and Sec20 (Frigerio, 1998; Kraynack *et al*, 2005; Ren *et al*, 2009; Meiringer *et al*, 2011). Moreover, the Dsl1 complex has also been connected to COPII-coated vesicles and their cargo, yet is an ER-resident complex (Andag *et al*, 2001; VanRheenen *et al*, 2001; Andag & Schmitt, 2003; Meiringer *et al*, 2011). We previously showed that Ykt6 interacts with the Dsl1 complex, though we could not unveil the functional significance of this interaction (Meiringer *et al*, 2011). Thus, we hypothesized that the Dsl1 complex could be involved in delivering Ykt6 to the PAS via COPII-coated vesicles, as those have been functionally connected with PAS/phagophore organization and autophagosome formation (Jensen & Schekman, 2011; Wang *et al*, 2014; Shima *et al*, 2019). The genes encoding for the Dsl1 complex subunits are essential as *YKT6*. Therefore, we took advantage of thermosensitive (*ts*) mutants, i.e., *dsl3-2* and *tip20-5*, for our analyses. To test whether the Dsl1 complex is required for autophagy, we monitored autophagy by assessing the delivery of GFP-tagged Atg8 inside FM4-64-stained vacuoles in these *ts* mutants under starvation at either permissive (24°C) or restrictive (37°C) temperature. As a control, we used the *ykt6-11 ts* strain (Gao *et al*, 2018b). As expected, we observed in these cells that most GFP-Atg8 was transported to the vacuole lumen at permissive temperature (Fig 3A). At restrictive temperature, however, GFP-Atg8 accumulated in puncta proximal to the vacuole. Some of these puncta contained a lumen, indicative of closed autophagosomes (Fig 3A; Gao *et al*, 2018b). We observed the same phenotype in the *dsl3* and *tip20* mutants, and in all cases, we also observed ring-like structures next to the vacuole (Fig 3A) and impaired autophagy (Fig 3B). The same observation was obtained with a *ts* mutant of the ER SNARE Sec20, which is closely associated with the Dsl1 complex (Fig 3A) (Ren *et al*, 2009; Meiringer *et al*, 2011). Using a protease protection assay (Nair *et al*, 2011b), we confirmed that the autophagosomes accumulated

adjacently to the vacuole in all these mutants are closed, i.e., complete (Fig 3C). Thus, these data show that like Ykt6, the Dsl1 complex is required for autophagosome fusion with vacuoles.

We next tested whether the Dsl1 complex is required for Ykt6 targeting to the PAS. In addition to the two mutants affected in the Dsl1 complex function (i.e., *tip20* and *dsl3*), we also used mutants impairing the biogenesis of COPII-coated vesicles, *sec12-1* (Barlowe & Schekman, 1993), to evaluate the contribution of each of them in Ykt6 localization to the PAS. As the Atg1 kinase complex is required for Ykt6 targeting to the PAS (Fig 2A and B), we first tested whether Atg1 localizes to the PAS under COPII-coat-inactivating conditions. Indeed, Atg1-positive puncta colocalized with the PAS marker protein GFP-Atg8 in the *sec12 ts* mutant at both permissive and restrictive temperatures (Fig 3D). This suggests that Atg1 remains at the PAS after inactivation of COPII-coated vesicles, in agreement with the notion that the Atg1 kinase complex forms an initial PAS precursor through phase separation rather than membrane delivery (Fujioka *et al*, 2020). In addition, GFP-Ykt6 colocalized with Dsl1-3xmCherry only under starvation conditions (Fig 3E and F). To test whether the Dsl1 complex travels with Ykt6 to the phagophore, we colocalized GFP-tagged Dsl3 and mCherry-tagged Atg8 under nutrient-rich and starvation conditions. In both cases, we did not observe colocalization between Dsl3 and Ykt6 (Fig 3G). This result suggests that Dsl1 complex interacts with Ykt6 at the ER, but does not accompany Ykt6 to the phagophore.

To further test whether COPII-coated vesicles and the Dsl1 complex contribute to the targeting of Ykt6 to the PAS, we colocalized GFP-Ykt6 with mCherry-Atg8 in *sec12*, *tip20*, and *dsl3* mutants at either permissive or restrictive temperature. In all cases, we observed impaired autophagy progression at restrictive temperature, as mCherry-Atg8 was not delivered into the vacuole, and Ykt6 accumulated in puncta that did not colocalize with Atg8 (Fig 4A and B). We noticed that *sec12 ts* cells exclusively accumulated Atg8 puncta and not ring-like structures as seen in *dsl* mutants (Fig 4A and B). This observation suggests that the disruption of general trafficking between ER and Golgi perturbs the overall biogenesis of autophagosomes at an early stage, confirming previous results (Jensen & Schekman, 2011; Graef *et al*, 2013; Suzuki *et al*, 2013; Wang *et al*, 2013; Shima *et al*, 2019).

As the Dsl1 complex is required for retrograde vesicular transport from the Golgi to the ER, we asked whether the accumulated Ykt6 puncta that we observed in the *tip20 ts* mutant are present on the

ER or COPII vesicles. To test this, we colocalized GFP-tagged Ykt6 with 3xmCherry-tagged Sec63 (a marker protein of the ER) and Sec13 (a subunit of the COPII coat) in *tip20 ts* mutant at either permissive or restrictive temperature. Importantly, we observed that the overall morphology of the ER was unaffected at both

temperatures. At permissive temperature, Ykt6 puncta colocalized with Sec63 and Sec13, but this colocalization was lost at the restrictive temperature (Fig 4C). Taken together, these data suggest that the Dsl1 complex is required for the delivery of Ykt6, possibly through COPII vesicles, to the phagophore. Thus, the phagophore

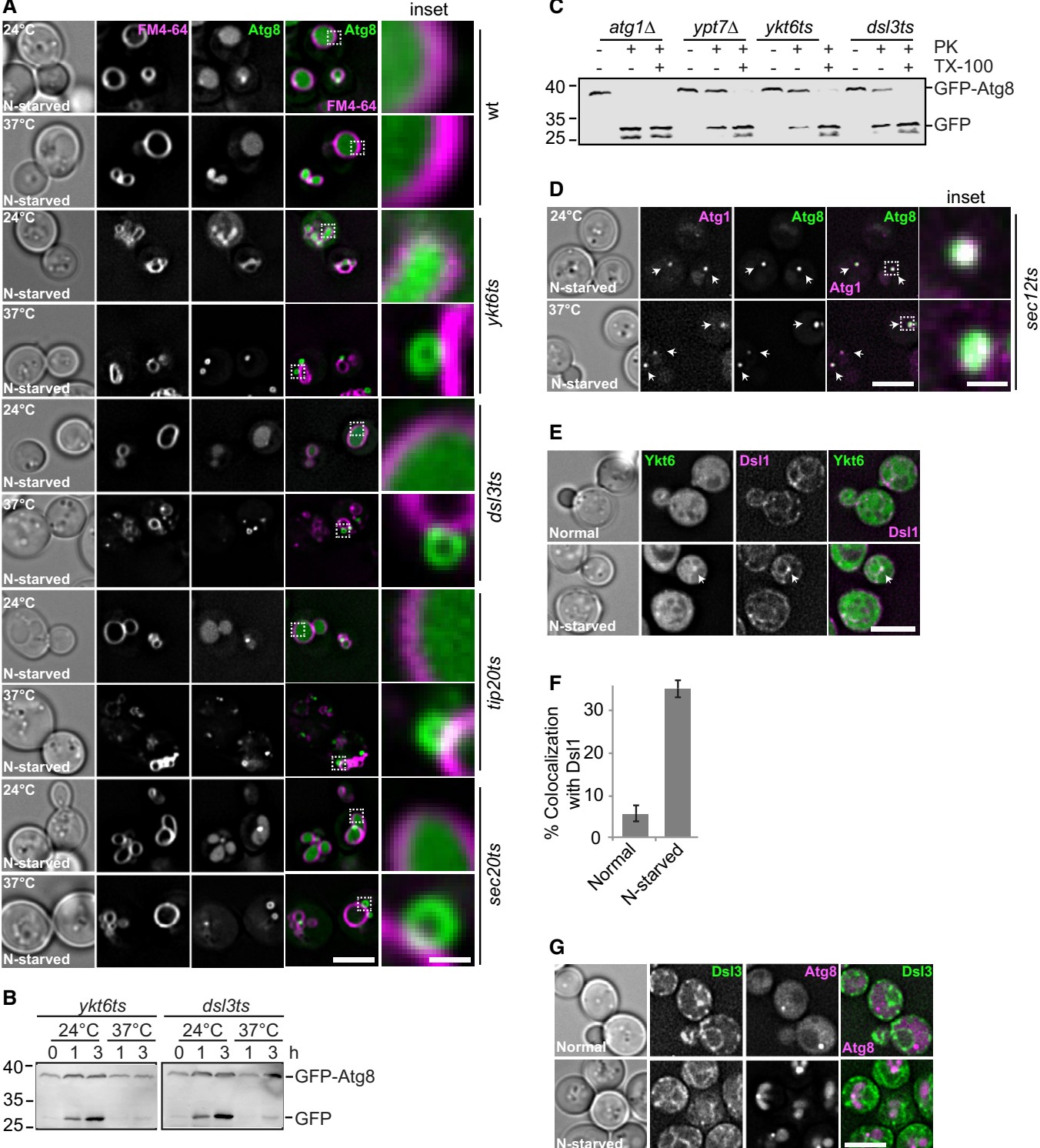

**Figure 3.**

**Figure 3. The Dsl1 complex is required for autophagy.**

A  Effect of *ykt6* and *dsl1* complex *ts* mutants on autophagosome–vacuole fusion. Cells carrying a CEN plasmid expressing GFP-Atg8 were grown at 24°C in SGC and then shifted to SG-N for 2 h at 24°C or 37°C. Vacuoles were stained by FM4-64 and analyzed by fluorescence microscopy. Scale bar, 5 μm. Scale bar of inset is 0.6 μm.

B  Analysis of autophagy over time. Cells were grown at 24°C and shifted to starvation medium at 24°C or 37°C for the indicated time periods. The cells were harvested, and proteins were extracted from cells by TCA precipitation. The extracted proteins were analyzed by SDS–PAGE and Western blotting against GFP.

C  Protease protection assay of *ykt6-11* and *dsl3* mutants. Lysates of *atg1Δ, ypt7Δ, ykt6ts,* and *dsl3ts* cells carrying a CEN plasmid expressing GFP-Atg8 were grown at 37°C and subjected to the protease (PK) protection (see Materials and Methods for details).

D  Localization of Atg8 relative to Atg1 during nitrogen starvation condition in *sec12-1 ts* mutant. Cells carrying a CEN plasmid expressing GFP-Atg8 and encoding 3xmCherry-tagged Atg1 were grown at 24°C in SDC and then shifted to SD-N for 2 h at 24°C or 37°C, and analyzed by fluorescence microscopy. Individual slices are shown. Scale bar, 5 μm. Scale bar of inset is 0.6 μm.

E  Localization of Ykt6 relative to Dsl1 during growth and nitrogen starvation conditions. Cells expressing GFP-tagged Ykt6 under the control of the *PHO5* promoter and genomically 3xmCherry-tagged Dsl1 were grown in SDC or SD-N for 2 h and analyzed by fluorescence microscopy. Individual slices are shown. Scale bar, 5 μm.

F  Percentage of Ykt6 puncta colocalizing with Dsl1. The data were quantified from (E). Error bars represent standard deviation of three independent experiments. Cells (*n* ≥ 250) and Dsl1 dots (*n* ≥ 50) were quantified.

G  Localization of Dsl3 relative to Atg8 during growth and nitrogen starvation. Cells expressing GFP-tagged Dsl3 and mCherry-tagged Atg8 were grown in SDC or SD-N for 2 h and analyzed by fluorescence microscopy. Scale bar, 5 μm.

Data information: Arrows in images indicate dot colocalization of Atg1 with Atg8 (D) and Ykt6 with Dsl1 (E).

can still expand in the absence of Ykt6, indicating that this SNARE is not required for the formation of this autophagosomal precursor.

To directly assess whether autophagosomes that accumulated in *dsl* mutants are defective in fusing with the vacuole, we took advantage of our *in vitro* fusion assay (Gao *et al*, 2018b). In this assay, GFP-Atg8-positive autophagosomes are purified from a *vam3Δ* strain, in which autophagosomes cannot fuse with the vacuole, before to monitor their fusion with purified vacuoles carrying Vac8-mCherry on their surface (Bas *et al*, 2018; Gao *et al*, 2018b). Fusion is assessed by measuring the GFP signal in the vacuole lumen. In a previous study, we showed that autophagosomes purified from a *ykt6 ts* mutant are fusion-incompetent (Gao *et al*, 2018b). To test whether *dsl* mutant-derived autophagosomes are also fusion-incompetent, we purified them from either *vam3Δ* or *dsl3* cells carrying GFP-Atg8. We then tested their fusion with vacuoles carrying Vac8-mCherry as mentioned (Fig 4D). Upon incubating autophagosomes derived from *vam3Δ* cells with vacuoles at 26°C in the presence of ATP, autophagosomes (as green dots) accumulate proximal to vacuoles (red circle) and eventually fuse, which results in a GFP signal within the vacuole lumen (Gao *et al*, 2018b). Importantly, fusion was blocked when autophagosomes from the *dsl3* mutant were assessed in this *in vitro* assay (Fig 4D and E). Our result is thus consistent with the notion that the Dsl1 complex is important for targeting Ykt6 to the PAS and consequently subsequent fusion of autophagosomes with the vacuole.

## Atg9 and COPII vesicles contribute independently to autophagosome formation

Having identified a contribution of the Dsl1 complex to target Ykt6 to autophagosomes, we wondered whether the Dsl1 complex also affects Atg9-containing vesicle delivery to the PAS. After induction of autophagy, Atg9-positive vesicles move from the Atg9 reservoirs, a post-Golgi compartment, and translocate to the PAS to contribute to phagophore formation (Mari & Reggiori, 2007; Reggiori & Tooze, 2012; Yamamoto *et al*, 2012). Recent studies showed that COPII-coated vesicles also serve as a membrane source for autophagosomes (Jensen & Schekman, 2011; Wang *et al*, 2014; Shima *et al*, 2019). Although Atg9 is proximal to COPII-coated vesicles during autophagy (Lynch-Day *et al*, 2010;

Graef *et al*, 2013), it is still only partially resolved how both Atg9-containing and COPII-coated vesicles contribute to the biogenesis of autophagosomes (Sanchez-Wandelmer *et al*, 2015).

We therefore quantified the number and size of autophagosomal intermediates that accumulate in different *ts* mutants, i.e., *ykt6, sec12, dsl3, tip20, vps11-1,* and *vam3*, and used the *atg1Δ* mutant as the negative control. The *vps11-1* cells are defective for all HOPS functions (Peterson & Emr, 2001) and accumulate a large number of Atg8-positive autophagosomes proximal to the vacuole at 37°C (Gao *et al*, 2018b). In contrast, *atg1Δ, ykt6, sec12, dsl3,* and *tip20* mutants had significantly less puncta of similar size than *vps11* and *vam3* mutants. Among those, only the *atg1Δ* and *sec12* cells had smaller puncta (Fig 5A and B). These data suggest that inactivation of either the Dsl1 complex or Ykt6 causes a decrease in autophagosome abundance, whereas the COPII-coated vesicles are required for the formation of autophagosomes. It is known that the interactions between COPII-coat and Atg9 vesicles increase the abundance of autophagosomes upon starvation, but not their size (Davis *et al*, 2016). It is thus possible that *ykt6 ts* and *dsl ts* mutants influence the communication of those vesicles, resulting in less autophagosomes.

We then asked whether the Dsl1 complex is needed for Atg9 targeting to the PAS. To test this, we colocalized Atg9-mCherry relative to GFP-Atg8 in the *dsl3* mutant at either the permissive temperature or the restrictive temperature. At both temperatures, Atg9-mCherry arrived at the PAS marked by GFP-Atg8 (Fig 5C). Furthermore, we observed Atg9-mCherry on the edge of the phagophore of giant BFP-Ape1 vesicles in the *dsl3* mutant regardless of the temperature (Fig 5D), indicating that this protein has normal distribution on this autophagosomal intermediate (Graef *et al*, 2013; Suzuki *et al*, 2013). Taken all together, these data agree with a model in that the recruitment of Atg9 to the PAS is independent of the Dsl1 complex.

## The Atg1 kinase complex directly regulates Ykt6 during autophagy

It is important that only complete autophagosomes fuse with vacuoles (Nakamura & Yoshimori, 2017). As Ykt6 is recruited to the PAS already during phagophore nucleation and expansion, we

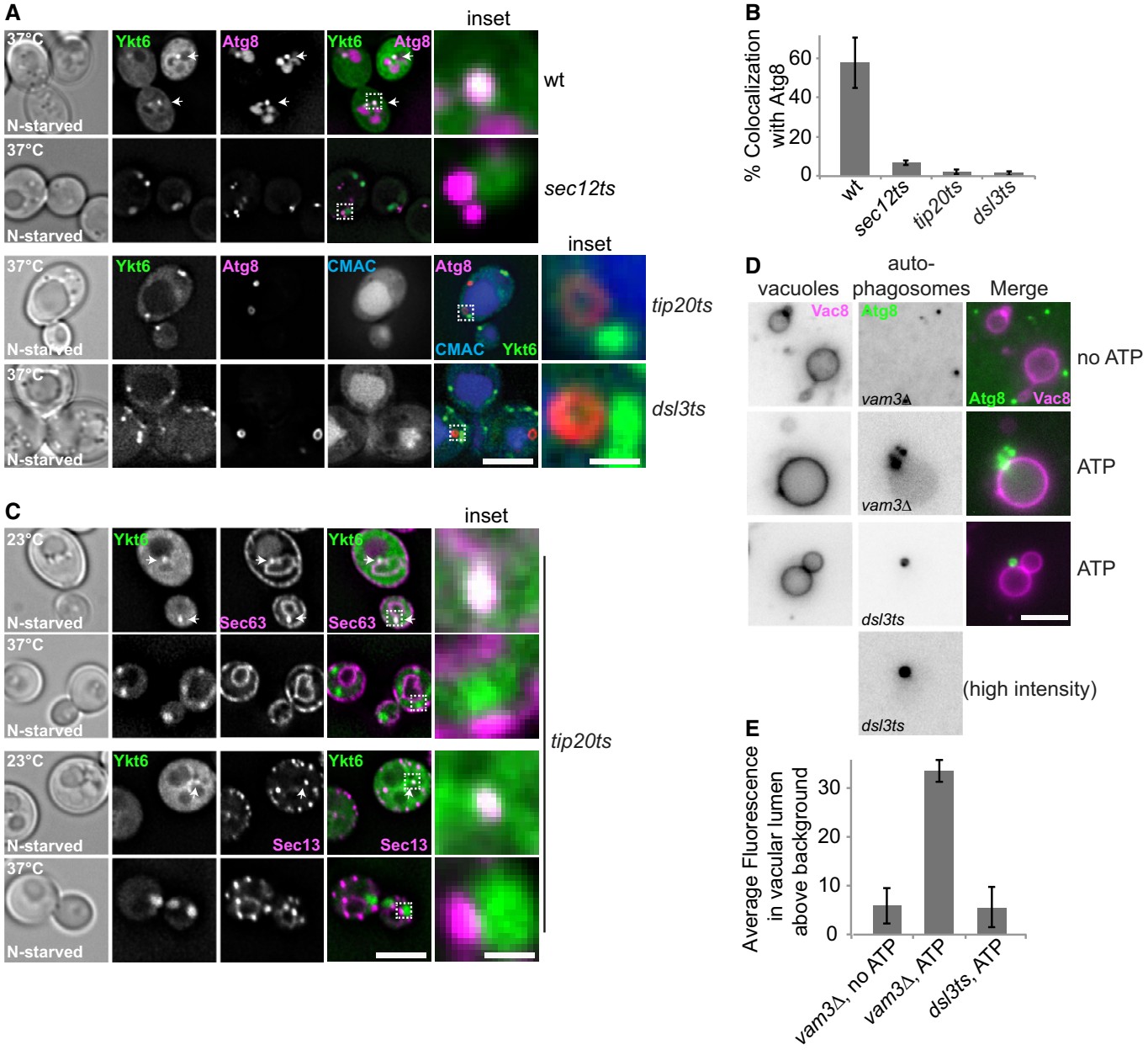

**Figure 4. Ykt6 targeting to the PAS requires the Dsl1 complex.**

A   Effect of Dsl and COPII *ts* mutants on Ykt6 localization to autophagosomes. Wild-type and selected *ts* strains (*sec12-1*, *tip20-5*, and *dsl3-2*) expressing GFP-tagged Ykt6 and mCherry-tagged Atg8 were grown at 24°C in SGC for 8 h and then shifted to SG-N for 2 h at 37°C. Vacuoles were stained by CMAC and analyzed by fluorescence microscopy. Scale bar, 5 μm. Scale bar in inset, 0.7 μm.

B   Percentage of Ykt6 puncta colocalizing with Atg8. The data were quantified from (A). Error bars represent standard deviation of three independent experiments. Cells (*n* ≥ 150) and Atg8 dots (*n* ≥ 100) were quantified.

C   Localization of Ykt6 relative to Sec63 and Sec13 during growth and nitrogen starvation conditions. *tip20-5* cells expressing GFP-tagged Ykt6 and 3xmCherry-tagged Sec63 or Sec13 were grown at 24°C in SGC for 8 h and then shifted to SG-N for 2 h at 24°C or 37°C, and analyzed by fluorescence microscopy. Scale bar, 5 μm. Scale bar of inset, 0.7 μm.

D   Effect of *dsl* mutation on autophagosome–vacuole fusion *in vitro*. The *vam3Δ* or *dsl ts* strain expressing Atg9-3xFLAG and GFP-Atg8 was starved for 3 h at 30°C before purifying autophagosomes, which was sufficient to induce the *ts* phenotype. Vacuoles were isolated from *pep4Δ* cells expressing Vac8-3xmCherry and then incubated with autophagosomes at 26°C for 1 h with or without ATP (Gao et al, 2018b). Scale bar, 5 μm.

E   Quantification of (D). Error bars represent standard deviation of 3 independent experiments. Fluorescence intensity of GFP-Atg8 in the vacuolar lumen was quantified by ImageJ using the ROI (region of interest) manager tool. Vacuoles (*n* ≥ 40) for each experiment were quantified.

Data information: Arrows indicate dot colocalization of Ykt6 and Atg8 (A) and Ykt6 with Sec63 (C).

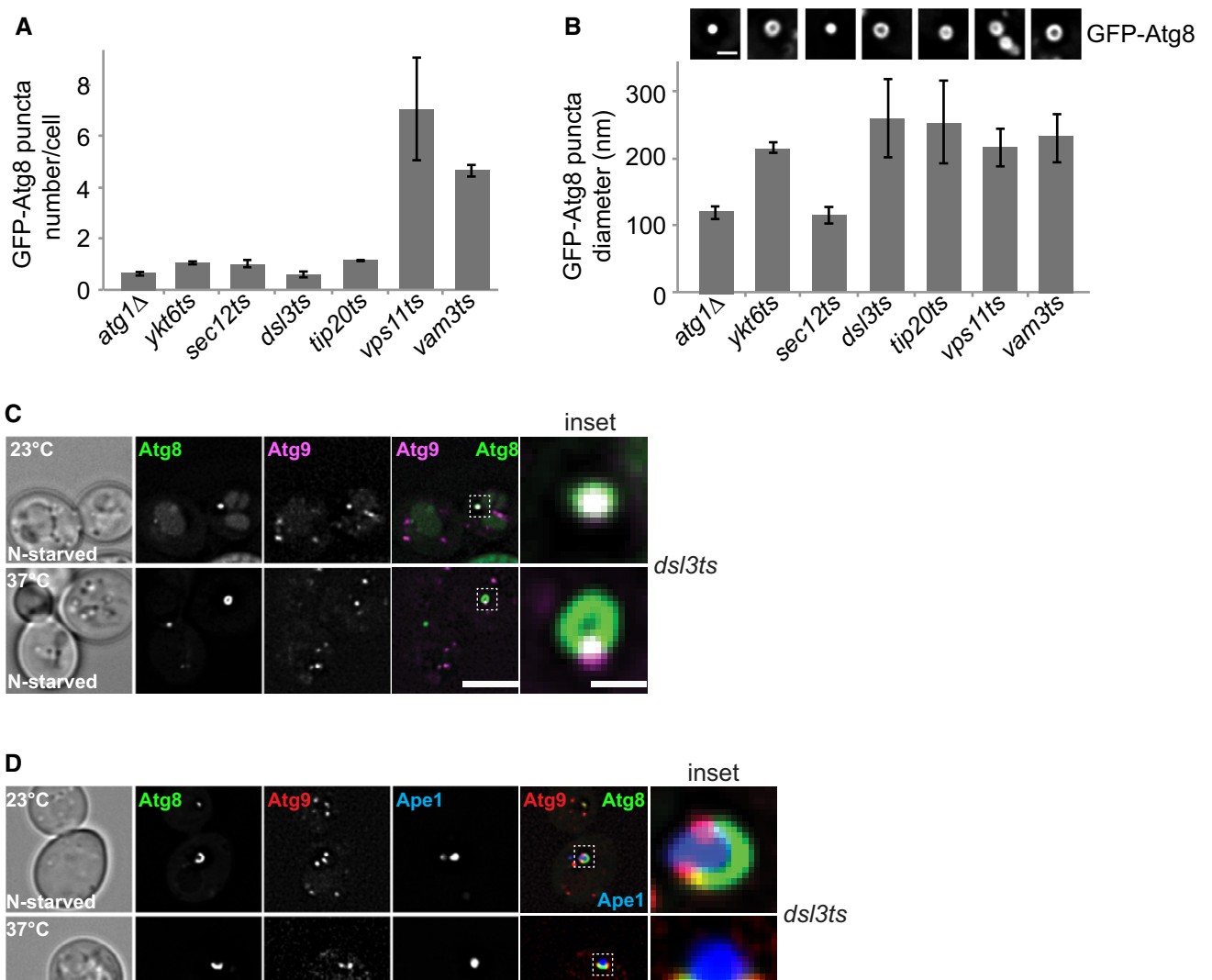

**Figure 5. Atg9 and the Dsl1 complex independently contribute to the autophagosome biogenesis.**

A, B  Quantification of size and number of autophagosomes in the indicated *ts* mutant strains. Error bars represent standard deviation of 3 independent experiments. GFP-tagged Atg8 in the different mutants were grown at 23°C (*ts* mutant) or 30°C (*atg1Δ*) in SDC and then shifted to SD-N for 2 h at 30°C (*atg1Δ*) or 37°C (*ts* mutant), and analyzed by fluorescence microscopy. Scale bar, 1 μm.

C  Localization of Atg8 relative to Atg9 during nitrogen starvation. *dsl3* cells expressing mCherry-tagged Atg9 and carrying a CEN plasmid expressing GFP-Atg8 were grown at 23°C in SDC and then shifted to SD-N for 2 h at 24°C or 37°C. Cells were analyzed by fluorescence microscopy. Scale bar, 5 μm. Scale bar for inset is 0.6 μm.

D  Atg9 localization in *dsl3* cells during starvation. *dsl3 ts* cells expressing mCherry-tagged Atg9 and carrying a CEN plasmid expressing GFP-Atg8 and plasmid pRS315-*CUP1pr-BFP-APE1* were grown in SDC medium containing CuSO₄ to induce formation of the giant Ape1 structure and starved for 1 h. Cells were analyzed by fluorescence microscopy. Scale bar, 5 μm. Scale bar for inset is 0.6 μm.

hypothesized the existence of a mechanism that either inhibits Ykt6 fusogenic function until autophagosomes are completed, or engage Ykt6 when those vesicles are formed. The Atg1 kinase complex, which in yeast is composed of Atg1, Atg13, Atg17, Atg29, and Atg31 (Chew *et al*, 2015), plays an essential role in autophagy. It is the most upstream factor of the Atg machinery and has a key role in initiating autophagosome formation (Papinski *et al*, 2014; Noda & Fujioka, 2015; Fujioka *et al*, 2020). Several lines of evidences have

shown that Atg1-mediated phosphorylation coordinates several steps that underlie the formation of an autophagosome. Self-phosphorylation and phosphorylation of Atg9 and Atg14 are essential to initiate the autophagosome biogenesis (Papinski *et al*, 2014; Wold *et al*, 2016). Interestingly, Atg1 phosphorylation of Atg4 keeps this protease inactive during the phagophore expansion to protect lipidated Atg8 from a premature release from autophagosomal membranes (Sánchez-Wandelmer *et al*, 2017). Upon autophagosome

completion, inactivation of Atg1, either by dissociation from autophagosomal membranes or by sequestration into the autophagosome lumen (Kraft *et al*, 2012), leads to the dephosphorylation of Atg4 and the concomitant release of Atg8, a step that is essential for the subsequent fusion of autophagosomes with vacuoles (Sánchez-Wandelmer *et al*, 2017).

Based on these considerations, we explored the possibility of the Atg1 kinase complex being involved in Ykt6 function regulation. We noticed that Ykt6 proteins from different species have a conserved putative Atg1 phosphorylation site within their SNARE domain, which corresponds to S182 in yeast Ykt6 (Fig 6A, see below). To ask whether Ykt6 and Atg1 are at the same structure, we colocalized GFP-Ykt6 relative to Atg1-3xmCherry in growing and nitrogen starvation conditions, and observed that both are present in the same puncta after nitrogen deprivation (Fig 6B). To test whether Ykt6 is a direct target of the Atg1 kinase complex, we then purified the Atg1-Atg13 subcomplex from yeast and performed an *in vitro* kinase assay with purified Ykt6 (Fig 6C). ATP-dependent activity of the Atg1–Atg13 subcomplex was apparent from the size shift of the kinase complex due to self-phosphorylation, whereas Ykt6 did not show a band shift. We therefore determined the possible phosphorylation of Ykt6 by protein mass spectrometry and identified four sites within the SNARE domain of Ykt6 that were changed with this post-translational modification: T151, T158, S182, and S183. The consensus amino acid sequence for Atg1 phosphorylation prefers aliphatic residues at position −3, especially leucine residues, and aliphatic and aromatic residues at positions +1 and +2 (Papinski *et al*, 2014). Residues L179 and S183 are highly conserved across species, yet S182 corresponds to the more likely target site of Atg1 (Fig 6A). Importantly, a recent *in vivo* study aiming at identifying Atg1 substrates confirmed that S182 and S183 of Ykt6 are indeed phosphorylated by Atg1 (Hu *et al*, 2019).

We first asked whether the recruitment of Ykt6 to autophagic structures is regulated by the Atg1–13 subcomplex. To test this, we colocalized GFP-tagged Ykt6 and mCherry-tagged Atg8 under nutrient-rich and starvation conditions in the wild type and cells expressing the kinase dead mutant *atg1*[D211A]. We observed that Ykt6 was recruited to the PAS only in wild-type cells after starvation (Fig 6D and E). These data indicate that the kinase activity of Atg1 may be important to target Ykt6 to the PAS. To further test whether Atg1 kinase complex also regulates the fusogenic activity of Ykt6 on autophagosomes, we generated expression plasmids encoding S182/S183 and T151/T158 as double point mutants with either non-phosphorylable alanine or phosphomimetic aspartate mutations, and transformed *ykt6Δ* cells carrying a plasmid encoding for wild-type Ykt6. When grown on 5-FOA to remove the plasmid expressing wild-type Ykt6, cells carrying the plasmid encoding for Ykt6[T151A,T158A] or Ykt6[T151D,T158D] were viable. Likewise, double mutant Ykt6[S182A,S183A] complemented the deletion, whereas the phosphomimetic Ykt6[S182D,S183D] variant died under these conditions (Fig 7A). As Ykt6 is essential, mutants altering the SNARE domain function likely affect all trafficking pathways that involve this protein, thus causing lethality.

As phosphomimetic mutants cannot be analyzed due to the loss of viability, we decided to analyze autophagy with GFP-tagged Atg8 in *ykt6Δ* cells carrying a plasmid encoding either for wild-type Ykt6 or for overexpressed Ykt6 phosphomutants, without growing them on 5-FOA plates to remove wild-type Ykt6. While cells expressing non-phosphorylatable Ykt6[S182A,S183A] did not impair GFP-Atg8 transport to the vacuole lumen, the ones carrying phosphomimetic Ykt6[S182D,S183D] strongly accumulated Atg8-positive dots proximal to the vacuole, indicative of a block of autophagosome fusion with the vacuole (Fig 7B and C).

To directly assess whether the phosphorylation keeps Ykt6 inactive until needed, we performed an *in vitro* autophagosome–vacuole fusion assay (Gao *et al*, 2018b) in the presence of the purified Atg1–Atg13 subcomplex and ATP. For this, we purified autophagosomes from cells expressing either wild-type Ykt6 or the non-phosphorylatable Ykt6[S182A,S183A], and observed efficient fusion of both types of autophagosomes with the isolated vacuoles (Fig 7D and E). Importantly, addition of the purified Atg1–Atg13 subcomplex strongly inhibited fusion in the presence of ATP, when wild-type Ykt6 was present on autophagosomes, but did not block fusion of autophagosomes carrying Ykt6[S182A,S183A] (Fig 7D and E). These results demonstrate that the Atg1 kinase complex regulates the fusogenic activity of Ykt6 on autophagosomes.

Overall, our data agree with a model where Atg1 inhibits the fusogenic activity of the SNARE Ykt6 during early steps of autophagosome formation, which precede the fusion with vacuoles.

## Discussion

Previous studies showed that R-SNARE Ykt6 is present on both yeast and metazoan autophagosomes, and it is required for their fusion with lysosomes/vacuoles (Bas *et al*, 2018; Gao *et al*, 2018b; Kriegenburg *et al*, 2019; Mizushima *et al*, 2019), yet how Ykt6 is specifically targeted to the surface of autophagosomes is still unknown. Here, we have uncovered the molecular determinants mediating the recruitment of Ykt6 to autophagosomal membranes. Our data support a model where Ykt6 interaction with the ER-resident Dsl1 complex is a prerequisite for its targeting possibly via COPII-coated vesicles to the PAS at the very early stage of autophagosome biogenesis (Fig 7F). Ykt6 is modified by dual prenylation (Shirakawa *et al*, 2020), and subsequently is possibly targeted to the ER, from where Ykt6 traffics via COPII-coated vesicles to the Golgi. At the PAS, the Atg1 kinase complex then regulates the fusogenic function of Ykt6, which is likely coordinated with the recruitment of Ypt7 and the HOPS complex. Our data agree with previous studies (Jensen & Schekman, 2011; Wang *et al*, 2014; Shima *et al*, 2019) and as discussed in Sanchez-Wandelmer *et al* (2015) imply that COPII-coated vesicles, which provide Ykt6, and Golgi-derived Atg9 vesicles both contribute to the phagophore nucleation (Fig 7F).

The highly conserved Ykt6 is essential for different transport pathways, including autophagy, intra-Golgi trafficking, Golgi to endosome transport, and vacuole–vacuole homotypic fusion (McNew *et al*, 1997; Ungermann *et al*, 1999; Dilcher *et al*, 2001; Zhang & Hong, 2001; Liu & Barlowe, 2002; Xu *et al*, 2002; Kweon *et al*, 2003; Fukasawa *et al*, 2004; Bas *et al*, 2018; Gao *et al*, 2018b; Matsui *et al*, 2018). As Ykt6 localizes to the ER, endosomes, vacuole, and Golgi under starvation conditions, Ykt6 may be transported to the PAS on vesicles from all these organelles and may even direct membranes to the growing phagophore. Our observations under autophagy-inducing conditions, however, show that Ykt6 arrives at the early stage of PAS assembly, prior to autophagosome completion (Fig 2). Consistently, the deletion of most core

ATG gene does not affect this redistribution of Ykt6 as large part of them is important for the expansion of the phagophore and not its nucleation (Nakatogawa et al, 2007; Obara et al, 2008; Zhong et al, 2009). Ykt6 is not present on the PAS in the mutant impaired in COPII-coated vesicle formation (Fig 4). These observations agree with our working model that COPII-coated vesicles carry Ykt6 to the nascent autophagosome. These vesicles, such as Atg9 vesicles, are then required for the phagophore formation and possibly contribute to its expansion as shown before (Reggiori et al, 2004, 2005; Yen et al, 2007; Jensen & Schekman, 2011; Backues et al, 2014; Wang et al, 2014; Shima et al, 2019). Interestingly, the Atg1 kinase

complex also plays a key role is tethering/targeting COPII-coated vesicles to the PAS (Wang et al, 2014). Consistently with this notion, Ykt6 cannot be delivered to the PAS in mutant strains lacking components of the Atg1 kinase complex, even if a PAS is present in these cells (Suzuki et al, 2007).

Our in vitro and in vivo data reveal that the Dsl1 complex is required for the delivery of Ykt6 to autophagosomes, and thus may explain the previously observed interaction between the Dsl1 complex and Ykt6 (Meiringer et al, 2011). In both ykt6 and dsl3 ts mutants, autophagosomes are complete (Fig 3C), similar as shown before for the vps11 and ypt7 mutants (Gao et al, 2018a,b). Ykt6 is

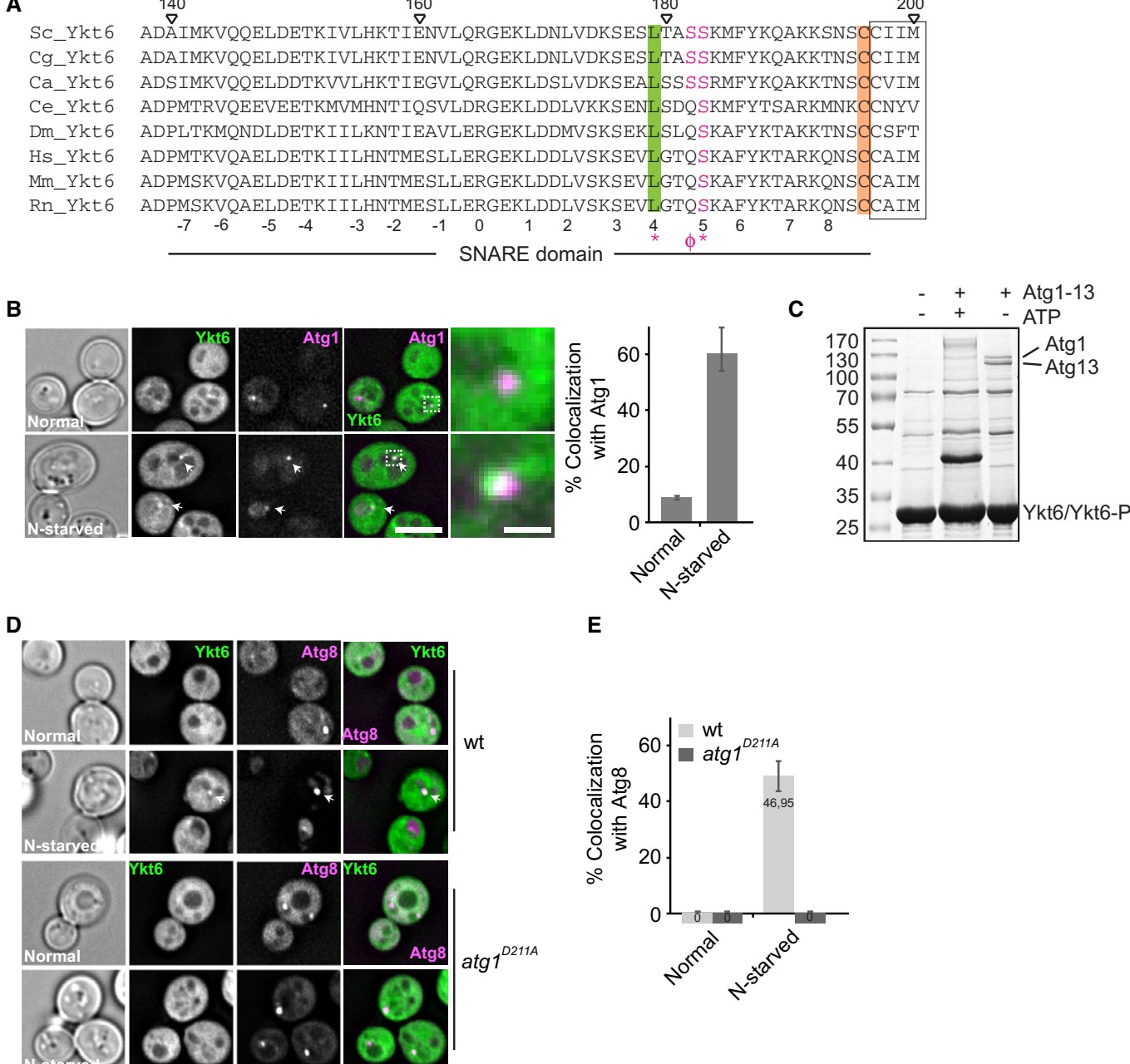

**Figure 6.**

**Figure 6. Ykt6 is a direct target of the Atg1 kinase complex during autophagy.**

A   Alignments of Ykt6 core domain with identified phosphorylation sites in SNARE domain. Sc, *Saccharomyces cerevisiae*; Cg, *Candida glabrata*; Ca, *Candida albicans*; Ce, *Caenorhabditis elegans*; Dm, *Drosophila melanogaster*; Hs, *Homo sapiens*; Mm, *Mus musculus*; Rn, *Rattus norvegicus*. Red symbols refer to consensus sequence of Atg1 kinase (Papinski *et al*, 2014). The conserved L is shown in green. The CaaX box is framed, and the conserved geranylgeranylated C next to the farnesylation site is marked in orange.

B   Localization of Ykt6 relative to Atg1 during growth and nitrogen starvation conditions. Cells expressing GFP-tagged Ykt6 under the control of the *PHO5* promoter and genomically 3xmCherry-tagged Atg1 were grown in SDC or SD-N for 2 h and analyzed by fluorescence microscopy. Individual slices are shown. Scale bar, 5 μm. Scale bar of inset is 0.6 μm. Right, percentage of Ykt6 puncta colocalizing with Atg1. Error bars represent standard deviation of three independent experiments. Cells ($n \geq 200$) and Atg1 dots ($n \geq 100$) were quantified.

C   *In vitro* phosphorylation of purified Ykt6 by the Atg1 kinase complex. Purified Ykt6 was incubated with the Atg1–Atg13 complex in the presence or absence of ATP, and Ykt6 band was cut out and analyzed by mass spectrometry.

D   Localization of Ykt6 relative to Atg8 under growth and nitrogen starvation conditions. Wild-type and *atg1*$^{D211A}$ cells expressing GFP-tagged Ykt6 under the control of the *PHO5* promoter and genomically Cherry-tagged Atg8 were grown in SDC or SD-N for 2 h and analyzed by fluorescence microscopy. Individual focal planes are shown. Scale bar, 5 μm.

E   Percentage of Ykt6 puncta colocalizing with Atg8. Error bars represent standard deviation of three independent experiments. Numbers in the column indicate the percentage. Cells ($n \geq 300$) and Atg8 dots ($n \geq 60$) were quantified.

Data information: Arrows indicate Ykt6-positive dots colocalizing with Atg1 (B) or Atg8 (D).

thus primarily required for the fusion of autophagosomes with vacuoles, but also contributes to the overall biogenesis, as there are much less autophagosomes in both *ykt6* and *dsl3 ts* mutants. The previous observation that Ykt6 can also block phagophore assembly and autophagosome closure (Nair *et al*, 2011a) could be due to the use of a stronger *ykt6* allele, which impairs all Ykt6 functions, including intra-Golgi transport. In contrast, the *ykt6-11* allele used here and before (Gao *et al*, 2018b) is far more specific as does not impair Golgi biogenesis and thus Atg9 vesicle formation, but only blocks fusion in compartments of the late endocytic pathway (Kweon *et al*, 2003). Disrupting the interactions of COPII and Atg9 vesicles decreases autophagosome abundance, but without affecting their size under starvation-induced conditions (Davis *et al*, 2016). Thus, it is possible that the *ykt6-11* mutant interferes with the fusion of these vesicles, resulting in less autophagosome of the same size. We speculate that the Dsl1 complex, which does not travel with Ykt6 to the PAS (Fig 3G), maintains a pool of Ykt6 at the ER membrane and makes it available for loading onto COPII-coated vesicles (Fig 7F).

In mammals, the autophagosomal SNARE SYNTAXIN17 is a substrate of the TBK1 kinase, which functions in the assembly of the mammalian PAS by controlling its interaction with the ULK complex, the equivalent of Atg1 complex in metazoan cells (Kumar *et al*, 2019). Importantly, our data show that the Atg1 kinase complex phosphorylates the SNARE domain of Ykt6 and regulates its fusogenic activity, while TBK1 controls SYNTAXIN17 sorting and its assembly with the ULK complex (Kumar *et al*, 2019). The Atg1 kinase complex may thus act similarly on the SNARE Ykt6 as it was previously shown for the Atg8-specific deconjugating enzyme Atg4, which recycled Atg8 from the complete autophagosome by removing its phosphatidylethanolamine anchor (Pengo *et al*, 2017; Sánchez-Wandelmer *et al*, 2017). Before closure of the autophagosome, the Atg1–kinase complex keeps Atg4 and Ykt6 inactive. Once Atg1 kinase complex is inhibited, dissociates, and/or is sequestered into the autophagosome (Kraft *et al*, 2012), Atg4 and Ykt6 get dephosphorylated and conduct their respective functions.

In conclusion, in this study we show that the Dsl1 complex and COPII-coated vesicles are determinants for the localization of the SNARE Ykt6 to the autophagosomal intermediates. Moreover, the identification of Atg1 as the critical kinase, which directly phosphorylates Ykt6 during autophagy, provides a novel example of how a SNARE protein can be regulated locally during organelle fusion. Future studies are needed to clarify how the entire fusion machinery is orchestrated during autophagy.

**Figure 7. The Atg1 kinase complex controls the fusogenic activity of Ykt6 during autophagy.**

A   Growth test of Ykt6 wild type and mutants. *ykt6Δ* cells carrying a *URA3* plasmid coding for wild-type Ykt6 and pRS413-*GAL1pr-YKT6-GFP* (wild type or mutant) were spotted as serial dilutions on SGC-HIS to select for the mutant plasmid or 5-fluoroorotic acid (5-FOA) to force the loss of the plasmid coding for wild-type Ykt6.

B   Effect of Ykt6 phosphomutants on autophagy. *ykt6Δ* cells expressing wild-type Ykt6 plus a pRS413 plasmid encoding *GAL1pr-YKT6* (wild type or mutants with S to A mutations [SA; S182, S183A] or S to D mutations [SD; S182D, S183D]) and a plasmid expressing GFP-Atg8 were grown in SGC and then shifted to SG-N for 2 h. Cells were analyzed by fluorescence microscopy. Individual slices are shown. Scale bar, 5 μm.

C   Quantification of Atg8 dots per cell from images in (B). Error bars represent standard deviation of three independent experiments. Cells ($n \geq 100$) and Atg8 dots ($n \geq 50$) were quantified.

D   Effect of Atg1-Atg13 subcomplex on autophagosome–vacuole fusion *in vitro*. The *vam3Δ* strain carrying pRS413 plasmid encoding *GAL1pr-YKT6* (wild type or mutant), and expressing Atg9-3xFLAG and GFP-Atg8, was starved for 3 h at 30°C before purifying autophagosomes. Vacuoles were isolated from *pep4Δ* cells expressing Vac8-3xmCherry and then incubated with autophagosomes in the presence or absence of 50 nM Atg1-Atg13 subcomplex at 26°C for 1 h with or without ATP (Gao *et al*, 2018b). Scale bar, 5 μm.

E   Quantification of (D). Error bars represent standard deviation of 3 independent experiments. Fluorescence intensity of GFP-Atg8 in the vacuolar lumen was quantified by ImageJ using the ROI (region of interest) manager tool. Vacuoles ($n \geq 40$) for each experiment were quantified.

F   Working model for the recruitment and regulation of Ykt6 during autophagy. Ykt6 arrives either after prenylation of via COPI-coated vesicles at the ER, where it binds the Dsl1 complex. Ykt6 is packaged into COPII-coated vesicles, which deliver Ykt6 independently from Atg9 vesicles to the PAS. Atg1–Atg13 kinase controls Ykt6 activity until autophagosomes are completely closed.

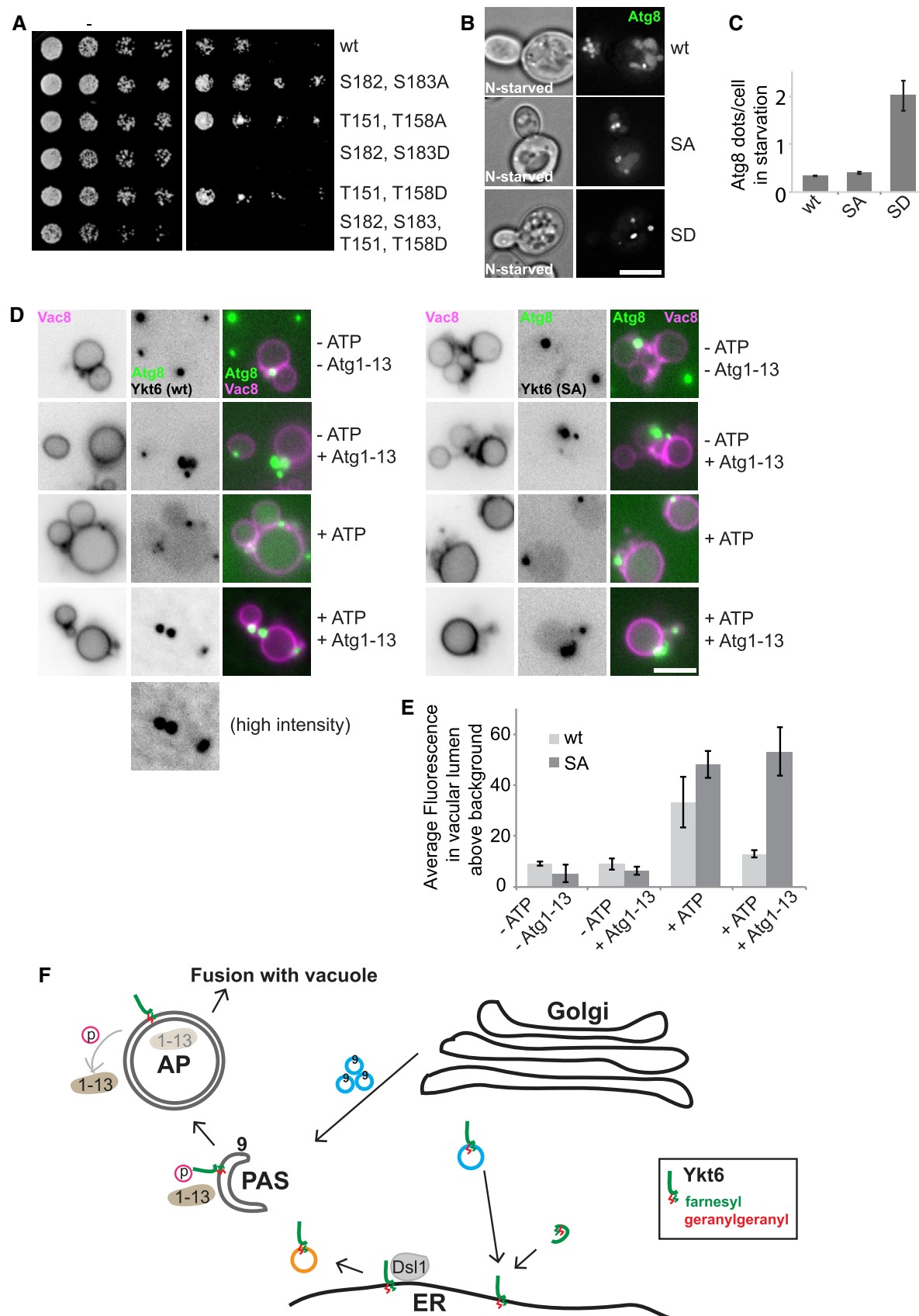

Figure 7.

# Materials and Methods

### Yeast strains and molecular biology

Strains used in this study are listed in Table EV1. Plasmids are listed in Table EV2. Deletions and tagging of genes were done by PCR-based homologous recombination with appropriate primers and template plasmids (Puig *et al*, 1998; Janke *et al*, 2004). Mutations on Ykt6 were carried out by QuikChange mutagenesis (Stratagene, La Jolla, CA). Mutations in Atg1 were generated by a CRISPR-Cas9 approach (Generoso *et al*, 2016).

### Preparation of protein extracts

Yeast cells were grown to a logarithmic (log) phase, pelleted, resuspended in ice-cold 13% trichloroacetic acid (TCA), incubated on ice for 30 min, and resuspended in ice-cold acetone by sonication. The final protein precipitate was resolubilized in SDS buffer (150 mM Tris–HCl, pH 6.8, 6 M Urea, 6% SDS, bromophenol blue, 10% β-mercaptoethanol) and boiled at 95°C for 5 min. Samples were finally analyzed by SDS–PAGE. GFP and Ape1 antibodies were used in these assays (Gao *et al*, 2018a).

### Light microscopy and image analysis

Yeast cells were cultured in synthetic complete medium (yeast nitrogen base without amino acids and with ammonium sulfate, 6.7 g/l) containing 2% glucose (SDC) or 2% galactose (SGC) to a log phase before to be transferred and starved in synthetic minimal medium lacking nitrogen and containing either 2% glucose (SD-N, 0.17% yeast nitrogen base without amino acids and ammonium sulfate, 2% glucose) or 2% galactose (SG-N, 0.17% yeast nitrogen base without amino acids and ammonium sulfate, 2% galactose) for 2 h. Cells or purified organelles were imaged on a DeltaVision Elite Imaging System based on an inverted microscopy, equipped with 100× NA 1.49 and 60× NA 1.40 objectives, a sCMOS camera (PCO, Kelheim, Germany), and an Insight SSI (TM) Illumination System. Stacks of 6–8 images with 0.2-μm spacing were collected before to be deconvolved using the SoftWoRx software (Applied Precision, Issaquah, WA). For purified autophagosomes and vacuoles, no Z stacks were recorded to avoid the bleaching of the fluorescent signal.

### Giant Ape1 assay

Yeast cells carrying the plasmid pRS315-*CUP1pr-BFP-APE1* (Torggler *et al*, 2016) were grown overnight in SDC medium lacking leucine and diluted to an early log phase next morning for 1 h before adding 250 μM CuSO$_4$ to induce the giant Ape1 oligomer formation during 4 h and subsequently starving cells in SD-N medium for 1 h to induce autophagy.

### Real-time 3D lattice light-sheet microscopy and image processing

Lattice light-sheet microscopy was performed on an exact home-built clone of the original design by the Eric Betzig group (Chen *et al*, 2014). Yeast cells expressing mCherry-Atg8 and carrying the plasmids pRS403-*GAL1pr-YKT6-GFP* (Meiringer *et al*, 2008; Gao *et al*,

2018b) and pRS315-*CUP1pr-BFP-APE1* were grown in SGC medium lacking leucine and switched to SG-N medium for 30 min as described for the Giant Ape1 assay. 1 OD$_{600}$ unit equivalents of cells were diluted in 40 μl of SG-N medium, and 5 μl of cells was spotted on the 5-mm round glass coverslips (Art. No. 11888372, Thermo Scientific) coated with concanavalin A for 5 min to make them adhere and then mounted on a sample holder specially designed for LLSM. The latter was inserted into a sample piezo placing the sample into the sample bath containing SG-N medium at room temperature (25°C). A three-channel image stack was acquired in sample scan mode by scanning the sample through a fixed light sheet with a step size of 500 nm which is equivalent to a ~ 271-nm slicing with respect to the *z*-axis considering the sample scan angle of 32.8°. We used a dithered square lattice pattern generated by multiple Bessel beams using an inner and outer numerical aperture of the excitation objective of 0.48 and 0.55, respectively. Each 3D image stack (512 × 320 × 150 voxels) contains 50–100 cells and was imaged at 50 frames per second in 9 s per volume. For time-lapse movies, we recorded every 30 s a full 3D stack for a total time of 30 min (60 time points). All three channels were sequentially excited using a 405-nm laser (LBX-405, Oxxius, Lannion, France) for BFP (Ape1), a 488-nm laser (2RU-VFL-P-300-488-B1R; MPB Communications Inc., Pointe-Claire, Canada) for GFP (YKT6), and a 560-nm laser (2RU-VFL-P-500-560-B1R; MPB Communications Inc.) for mCherry (ATG8). Fluorescence was detected by a sCMOS camera (ORCA-Flash 4.0, Hamamatsu, Japan) using an exposure time of 18.3 ms for each channel. The final pixel size in the image is 103.5 nm. The raw data were further processed by using an open-source LLSM post-processing utility called LLSpy (https://github.com/tlambert03/LLSpy) for deskewing, deconvolution, 3D stack rotation, and rescaling. Deconvolution was performed by using experimental point spread functions and is based on the Richardson–Lucy algorithm using 10 iterations. Finally, the image data were analyzed using the software Imaris 9.2 (Bitplane, Zurich, Switzerland).

### Protease protection assays

Cells carrying a centromeric plasmid expressing GFP-Atg8 were grown overnight in SDC medium lacking uracil, diluted to an OD$_{600}$ of 0.1–0.2 the next morning, and then grown to a mid-log phase before shifted into SD-N for 2 h to induce starvation. Six OD$_{600}$ equivalent unit of cells was harvested by centrifugation at 500 *g* for 5 min and washed once with 400 μl of DTT buffer (100 mM Tris, pH 9.4, 10 mM DTT). Cells were then resuspended in 2 ml of DTT buffer and incubated at 30°C shaking at 300 rpm for 15 min. After incubation, cells were harvested by centrifugation at 600 *g* for 6 min and resuspended with 600 μl of SP buffer (1 M sorbitol, 20 mM PIPES/KOH, pH 6.8) containing 0.3 mg/ml lyticase. The samples were incubated at 30°C for 25 min and mixed by inverting the tube during the incubation. Spheroplasts were collected by centrifugation at 2,000 *g* for 10 min and resuspended carefully in 600 μl of SP buffer. Spheroplasts were collected again by centrifugation at 2,000 *g* for 10 min and resuspended in 200 μl of lysis buffer PS200 (20 mM PIPES/KOH, pH 6.8, 200 mM sorbitol, 5 mM MgCl$_2$). The samples were incubated on ice for 5 min and inverted three times during the incubation. After incubation, the unbroken cells and cell debris were removed by centrifugation at 500 *g* for 10 min at 4°C. Supernatants were treated and mixed with proteinase K (PK;

40 μg/ml) in the absence or the presence of 0.4% Triton X-100 (TX), and incubated on ice for 20 min as previously described (Nair *et al*, 2011b). Following the PK treatment, proteins were precipitated with TCA for immunoblot analysis using anti-GFP.

## Isolation of autophagosomes

Autophagosomes were purified as described before (Gao *et al*, 2018b). Yeast cells expressing GFP-Atg8 and Atg9-3xFLAG were grown to approximately $OD_{600} = 1.0$ in 1 l of YPD medium overnight and switched to SD-N medium the next morning for 3 h to induce autophagy. Cells were harvested by centrifugation at 4,000 *g* for 3 min, resuspended with DTT buffer (0.1 M Tris–HCl, pH 9.4, 10 mM DTT), and incubated at 30°C water bath for 15 min. Cells were collected and resuspended with the spheroplasting buffer (0.16× SD-N, 0.6 M sorbitol, 50 mM KPi, pH 7.4) containing 0.3 mg/ml lyticase at 30°C for 30 min. Spheroplasts were collected by centrifugation at 1,000 *g* for 3 min and resuspended in 3 ml lysis buffer (0.2 M sorbitol, 50 mM KOAc, 2 mM EDTA, 20 mM HEPES/KOH pH 6.8) containing a protease inhibitor cocktail (PIC; 0.1 mg/ml of leupeptin, 1 mM o-phenanthroline, 0.5 mg/ml of pepstatin A, 0.1 mM Pefabloc), 1 mM PMSF, and 1 mM DTT. The samples were incubated with 0.4 mg/ml DEAE on ice for 5 min and heat-shocked for 2 min at 30°C. The supernatant was collected by centrifugation at 400 *g* for 10 min at 4°C and then centrifuged at 15,000 *g* for 15 min at 4°C. The pellet was resuspended with 1 ml lysis buffer as mentioned before, layered onto discontinuous iodixanol gradients (1.5 ml of 20%; 6 ml of 10%; 4 ml of 5%), and gradients were centrifuged at 100,000 *g* for 60 min at 4°C. Fractions were collected from the 10–20% interface (1 ml) and incubated with anti-FLAG beads (Sigma-Aldrich, Germany) overnight at 4°C. Beads were washed with 2 ml of ice-cold lysis buffer by centrifugation at 30 *g* for 2 min at 4°C. Finally, bound autophagosomes were eluted with lysis buffer containing 0.25 μg/μl FLAG peptide.

## Autophagosome–vacuole fusion assay

Autophagosome–vacuole fusion assay was performed as described before (Gao *et al*, 2018b). Yeast cells expressing Vac8-3xmCherry from *pep4Δ* strain for vacuole isolation were grown to approximately $OD_{600} = 1.0$ in 1 l YPD medium overnight and purified via Ficoll gradient centrifugation as previously described (Haas, 1995). The isolated autophagosomes and vacuoles were diluted with 0% Ficoll buffer (10 mM 1,4-piperazinediethanesulfonic acid [PIPES]/KOH, pH 6.8, 200 mM sorbitol, 0.1×PIC) to 1.5 mg/ml and 0.3 mg/ml, respectively. The fusion reaction (15 μg autophagosomes, 3 μg of vacuoles) was performed in the fusion buffer (125 mM KCl, 5 mM $MgCl_2$, 20 mM sorbitol, 1 mM PIPES/KOH, pH 6.8) with 10 μM CoA, 0.01 μg of 6xHis-Sec18, and an ATP-regenerating system (5 mM ATP, 1 mg/ml creatine kinase, 400 mM creatine phosphatase, 10 mM PIPES/KOH, pH 6.8, 0.2 M sorbitol) for 30 min at 26°C. Fusion efficiency was analyzed by fluorescence microscopy.

## Tandem affinity purification

Tandem affinity purification was performed as described (Gao *et al*, 2018a). The cells over expressing both Atg13 and Atg1-TAP were grown at 30°C to an $OD_{600}$ of 2–3, and cells were harvested and lysed in lysis buffer [PBS, 10% glycerol, 0.5% CHAPS, 1 mM PMSF, 0.05× protease inhibitor cocktail (1× = 0.1 mg/ml of leupeptin, 1 mM o-phenanthroline, 0.5 mg/ml of pepstatin A, 0.1 mM Pefabloc)] containing phosphatase inhibitors (1 mM NaF, 1 mM $Na_3VO_4$, 20 mM ß-glycerophosphate). Lysates were centrifuged at 100,000 *g* for 1 h, and the cleared supernatants were collected and incubated with IgG Sepharose beads (GE Healthcare) for 2 h at 4°C. After incubation, beads were collected by centrifugation at 800 *g* for 2 min at 4°C and washed with ice-cold 10–15 ml lysis buffer containing 0.5 mM DTT and 10% glycerol. Finally, bound proteins were eluted by TEV cleavage for 90 min at 4°C and part of the purified proteins were analyzed on SDS–PAGE. The purified Atg1–Atg13 subcomplex was pooled and dialyzed against assay buffer, which was changed twice to allow for addition to the *in vitro* autophagosome–vacuole fusion assay.

## Kinase activity assay

The purified Atg1–Atg13 kinase complex (0.5 μg/μl) and Ykt6 (3 μg/μl) were mixed with 20 μl of 2× phosphorylation buffer (200 mM NaCl, 10 mM Tris, pH 7.5, 10 mM $MgCl_2$, 0.4 mM EDTA, 10% glycerol) in the absence or the presence of 1 mM ATP. The samples were incubated at 30°C for 40 min and examined on SDS–PAGE after incubation. Ykt6 bands were cut from SDS–PAGE and analyzed by mass spectrometry for phosphorylated peptides.

## *Escherichia coli* protein expression and purification

Ykt6 was purified from *E. coli* BL21 (DE3) Rosetta cells. Protein expression was induced with 0.5 mM IPTG overnight at 16°C until 0.6 $OD_{600}$. Cells were lysed in lysis buffer (50 mM HEPES/NaOH, pH 7.5, 150 mM NaCl, 1 mM PMSF, 1× protease inhibitor cocktail (1× = 0.1 mg/ml of leupeptin, 1 mM o-phenanthroline, 0.5 mg/ml of pepstatin A, 0.1 mM Pefabloc)). The lysates were centrifuged at 30,000 *g* for 20 min at 4°C, and the cleared supernatant was collected and incubated with Ni-NTA beads for 1 h at 4°C. After incubation, beads were collected by centrifugation at 800 *g* for 2 min at 4°C and washed with ice-cold 25 ml lysis buffer containing 20 mM imidazole. Bound proteins were eluted with buffer containing 300 mM imidazole. The eluted protein was applied to a NAP-10 column (GE Healthcare) to achieve transfer to the following buffer (50 mM HEPES/NaOH, pH 7.4, 150 mM NaCl, 10% glycerol).

# Data availability

This study includes no data deposited in external repositories.

**Expanded View** for this article is available online.

# Acknowledgements

We thank Angela Perz and Kathrin Auffarth for expert technical assistance. F. Reggiori was supported by ALW Open Program (ALWOP.310), ZonMW TOP (91217002), an Open Competition ENW-KLEIN (OCENW.KLEIN.118), Marie Skłodowska-Curie Cofund (713660), and Marie Skłodowska-Curie ITN (765912) grants. C. Ungermann was supported by a grant of the Deutsche Forschungsgemeinschaft (UN111/7-3 and 13-1) and the Sonderforschungsbereich 944

(Project P11). R. Kurre and J. Piehler were supported by the Sonderforschungs-
bereich 944 (Projekt Z01) and the iBiOS funds of the DFG. Open access funding
enabled and organized by Projekt DEAL.

## Author contributions

JG, FR, and CU conceived and designed experiments. JG performed all experi-
ments. RK and JP supported JG in her analyses of the lattice light-sheet micro-
scope. JR purified the Atg1-Atg13 complex. SW did the mass spectrometry
work, and data were analyzed by FF. All authors analyzed the results. JG, FR,
and CU wrote the manuscript.

## Conflict of interest

The authors declare that they have no conflict of interest.

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
