## [Review Process File · EMBO Reports]

Function of the SNARE Ykt6 on autophagosomes requires the Dsl1 complex and the Atg1 kinase complex

Jieqiong Gao, Rainer Kurre, Jaqueline Rose, Stefan Walter, Florian Fröhlich, Jacob Piehler, Fulvio Reggiori, and Christian Ungermann

DOI: [10.15252/embr.202050733](https://doi.org/10.15252/embr.202050733)

Corresponding author(s): *Christian Ungermann (cu@uos.de)*

Review Timeline:	Submission Date:	24th Apr 20
	Editorial Decision:	5th Jun 20
	Revision Received:	4th Aug 20
	Editorial Decision:	1st Sep 20
	Revision Received:	4th Sep 20
	Accepted:	14th Sep 20

Transaction Report:

Dear Dr. Ungermann

Thank you for the submission of your research manuscript to our journal. I apologize for the delay in handling your manuscript, but we have only recently received the full set of referee reports that is copied below.

As you will see, the referees acknowledge that the findings are potentially interesting. However, referee 1, an expert in autophagy, and referee 2, an expert in intracellular trafficking, both point out several technical concerns and have a number of suggestions for how the study should be strengthened. The evidence that p-Ykt6 is non-fusogenic and that Ykt6 is indeed phosphorylated on the phagophore membrane should be strengthened, as outlined by referee 1. The role of Dsl-1 in targeting Ykt6 needs to be substantiated and more evidence is required that this role is direct and requires the interaction with Ykt6 as opposed to a more general disruption of the secretory pathway. The latter concern also relates to the role of COPII in this process. It is however not essential to identify a phosphatase for Ykt6, which might indeed be the focus of an independent project (ref 3, last point).

From these comments it is clear that a major revision will be required before publication in EMBO reports can be considered. Yet, given the constructive comments and the support from at least two referees, I would like to give you the opportunity to address the concerns and would be willing to consider a revised manuscript with the understanding that the referee concerns must be fully addressed and their suggestions taken on board as outlined above in their reports. Should you decide to embark on such a revision, acceptance of the manuscript will depend on a positive outcome of a second round of review. It is EMBO reports policy to allow a single round of revision only and acceptance or rejection of the manuscript will therefore depend on the completeness of your responses included in the next, final version of the manuscript.

We invite you to submit your manuscript within three months of a request for revision. This would be September 5th in your case. Having said so, given the current COVID-19 related lockdowns of laboratories, we have extended the revision time for all research manuscripts under our scooping protection to allow for the extra time required to address essential experimental issues. Please contact me if you wish to discuss the time needed and the revisions further (m.rembold@emboreports.org).

- 1) A data availability section is missing.
- 2) Your manuscript contains error bars based on $n=2$. Please use scatter blots showing the individual datapoints in these cases. The use of statistical tests needs to be justified.

2) individual production quality figure files as .eps, .tif, .jpg (one file per figure).

Please download our Figure Preparation Guidelines (figure preparation pdf) from our Author Guidelines pages

<https://www.embopress.org/page/journal/14693178/authorguide> for more info on how to prepare your figures.

4) a complete author checklist, which you can download from our author guidelines (). Please insert information in the checklist that is also reflected in the manuscript. The completed author checklist will also be part of the RPF.

5) Please note that all corresponding authors are required to supply an ORCID ID for their name upon submission of a revised manuscript (). Please find instructions on how to link your ORCID ID to your account in our manuscript tracking system in our Author guidelines

()

6) We replaced Supplementary Information with Expanded View (EV) Figures and Tables that are collapsible/expandable online. A maximum of 5 EV Figures can be typeset. EV Figures should be cited as 'Figure EV1, Figure EV2" etc... in the text and their respective legends should be included in the main text after the legends of regular figures.

7) Please note that a Data Availability section at the end of Materials and Methods is now mandatory. The Data Availability Section is restricted to new primary data that are part of this study.

In case you have no data that requires deposition in a public database, please state so instead of referring to the database.

See also < <https://www.embopress.org/page/journal/14693178/authorguide#dataavailability>>).

8) We would also encourage you to include the source data for figure panels that show essential data. Numerical data should be provided as individual .xls or .csv files (including a tab describing the data). For blots or microscopy, uncropped images should be submitted (using a zip archive if multiple images need to be supplied for one panel). Additional information on source data and instruction on how to label the files are available .

10) Regarding data quantification:

- Please ensure to specify the name of the statistical test used to generate error bars and P values, the number (n) of independent experiments underlying each data point (please define the nature of replicates, i.e., technical or biological), and the test used to calculate p-values in each figure legend. Discussion of statistical methodology can be reported in the materials and methods section, but figure legends should contain a basic description of n, P and the test applied.

IMPORTANT: Please note that error bars and statistical comparisons may only be applied to data obtained from at least three independent biological replicates. If the data rely on a smaller number of replicates, scatter blots showing individual data points are recommended.

- Graphs must include a description of the bars and the error bars (s.d., s.e.m.).

11) As part of the EMBO publication's Transparent Editorial Process, EMBO reports publishes online a Review Process File to accompany accepted manuscripts. This File will be published in conjunction with your paper and will include the referee reports, your point-by-point response and all pertinent correspondence relating to the manuscript.

I look forward to seeing a revised version of your manuscript when it is ready. Please let me know if you have questions or comments regarding the revision.

Yours sincerely,

Martina Rembold, PhD
Editor
EMBO reports

Referee #1:

In this report, the authors show that GFP-Ykt6 is recruited to the PAS and autophagosomes in an Atg1 and Dsl1 complex-dependent manner. Although Ykt6 is present on the phagophore, the

authors propose that its fusogenic activity is kept inhibited by phosphorylation of S182 and S183 by Atg1 until required. Accordingly, overexpression of the Ykt6S182D/S183D mutant blocks autophagosome-vacuole fusion.

Overall, this is solid work, and the data are clear and well-presented.

Major concerns

1. The authors hypothesize that phosphorylated Ykt6 is non-fusogenic, but the evidence for this is not strong. As the authors have developed a well-working in vitro fusion assay system, it would be possible to test the effect of Atg1-dependent phosphorylation of Ykt6 on autophagosomal fusion (by combining the experiments in Fig. 3H and Fig. 5C). Alternatively, if Ykt6S182A/S183A is constitutively active, it would be worth testing whether premature fusion of a phagophore with the vacuole occurs (e.g., by live-cell imaging of Ape1-overexpressing cells).

2. It is not demonstrated whether Ykt6 on the phagophore membrane is indeed phosphorylated. It would be ideal to determine by mass spectrometry whether phosphorylated Ykt6 is enriched in the phagophore or autophagosomal fractions in an Atg1-dependent manner.

3. The data in Fig. 3G is one of the most important data in this study. Some quantification is required (including a wild-type control). A wild-type control is also missing and should be included in Fig. 3A.

4. The data in Fig. 2A and B show that Atg17 is required for Ykt6 recruitment. Given that Atg17 is not required for the Cvt pathway, is Ykt6 dispensable for Cvt?

Minor points

Why is there less accumulation of autophagosomes in the absence of Ykt6 or the Dsl1 complex (Fig. 4B)? Some explanation is required.

P. 11: (Figure 3F, G) should be (Figure 3H, L). The same for its legend.

P. 16: (Figure 5G) should be (Figure 5F).

Referee #2:

In this study the authors build on their previous works identifying Ykt6 as an R-SNARE on autophagosomes. They demonstrate that Ykt6 localises to the autophagosome upon nitrogen starvation in yeast, as well as several other organelles, and that this occurs early during the initiation of autophagosome formation in a manner dependent on Atg1 protein kinase complex. Ykt6, as well as the ER-resident Dsl-1 complex are then shown to be required for autophagosome/lysosome fusion. The data for this are clear and well presented. Lead by their previous finding that Ykt6 can interact with Dsl-1, the authors then show that Dsl-1 and COPII are required for Ykt6 delivery to the phagophore. In Dsl-1 mutant cells, autophagosomes form but are fusion incompetent, and in the latter, autophagosome biogenesis is inhibited. Finally, the authors show that premature Ykt6-mediated fusion of the autophagosome with lysosomes/vacuoles is prevented by ATG1-dependent phosphorylation.

The strengths of this paper lie in the characterisation of Ykt6 recruitment to the phagophore and its regulation by phosphorylation to control the timing of fusion with the vacuole. This adds a key detail to current models of the fusion of interest to the autophagy field and those interested in

SNARE regulation. On the other hand, whilst Dsl-1 does seem to be required for Ykt6 localisation to the phagosome, the data describing a more specific role for Dsl-1 in targeting Ykt6 are compelling but indirect. The evidence for COPII's role in this process seems even more circumstantial. Thus this section does not progress knowledge much beyond what could already be deduced from the current literature, but if substantially improved would be more generally of interest to the membrane trafficking field. This would require a significant amount of work however which may not be feasible in the current crisis.

Major concerns

The authors propose that 'Our data support a model where Ykt6 interaction with the ER resident Dsl1 complex is a prerequisite for its targeting possibly via COPII coated vesicles to the PAS at the very early stage of autophagosome biogenesis'. The advancement here is that Dsl-1 is required for Ykt6 targeting to the autophagosome, since these interactions are already known. However, the evidence for this seems largely circumstantial and conclusions poorly supported as the data could be interpreted in a number of ways.

1. Whilst Dsl-1 activity is shown to be required, a requirement for a direct Dsl-1-Ykt6 interaction has not been demonstrated, for example with mutations selectively interrupting this interface.

2. Given the known role of Dsl-1 as an ER-resident membrane tether for ER-ER fusions and retrograde COPI transport, the requirement for Dsl-1 in Ykt6 targeting to the phagophore could easily be due to indirect effects on the structure of, and membrane flux through, the secretory pathway. This could explain why autophagosome abundance is reduced if membrane is less freely available or dynamic. Alternatively, Dsl-1 could be important for the recycling of machinery required to transport Ykt6 to the phagophore but not Ykt6 itself. The authors do not appear to have looked at the health of the secretory pathway in the absence of Dsl-1 or the structure of the ER/ERES, for example with secretion assays or EM of ER and Golgi structures.

3. Ykt6 on ERGIC membranes has been shown to be recruited to ERES by TANGO1 (Santos) - it is therefore possible that Dsl-1 is similarly required for the retrograde transport of Ykt6 back to the ER for re-routing to the phagophore. The authors have not looked to see whether Ykt6 is still localised to ER membranes in the Dsl-1 mutants which would help decipher whether Dsl-1 is required for general ER localisation of Ykt6 or more specifically involved in Ykt6 targeting to COPII vesicles destined for the phagophore.

4. Although an interaction between Ykt6 and Dsl-1 has been shown before, the authors do not investigate this directly here. The Ykt6-Dsl-1 interaction appears weaker than that between Dsl-1 and other SNAREs (Meiringer et al) so is it influenced by conditions that induce autophagy - for example is this interaction more prevalent following nitrogen starvation? Is Ykt6 modified on starvation so that the interaction is enhanced? Answers to these questions would demonstrate a more direct link and could provide insight into how Ykt6 is sent to the phagophore instead of the Golgi when autophagy is induced.

5. Autophagosomes from Dsl-1 mutants are fusion incompetent in in vitro assays, as are those from Ykt6 mutants, and thus the authors conclude Dsl-1 targets Ykt6 to autophagosomes for fusion. Whilst this is a logical explanation it is merely inferred from indirect evidence - a rescue by adding Ykt6 to the membranes from Dsl-1 mutants in this assay would demonstrate that it is Ykt6 targeting and not the targeting of some other machinery that is important. As a tether for retrograde ER trafficking, Dsl-1 could be affecting any number of things.

The authors also state 'Ykt6 is not present on the PAS in the mutant impaired in COPII- and COPI-coated vesicle formation (Figure 3). These observations agree with our working model that COPII-coated vesicles carry Ykt6 to the nascent autophagosome'.

6. This could again easily be due to general disruption to the secretory pathway. Indeed autophagosome biogenesis is disrupted in the COPII mutants implying much bigger defects that would indirectly affect the recruitment of machinery. Also Ykt6 could be coming from other organelles. So whilst this is a good working model caution should be applied in linking COPII to Ykt6 delivery without further evidence for example here 'In conclusion, in this study we show that the Dsl1 complex and COPII-coated vesicles are determinants for the localization of the SNARE Ykt6 to the autophagosomal intermediates' Demonstration of colocalisation between Ykt6 and COPII components, especially upon nitrogen starvation would help better support this model. In Figure 3E Dsl-1 and Ykt6 co-localise to a single puncta upon serum starvation - is this the phagophore? This would imply Dsl-1 travels with Ykt6. Alternatively this could be ER.

7. This should be investigated with double or triple fluorescence labels to determine where they coalesce as it could influence interpretation of models.

Minor concerns

8. In figure 5C it is difficult to really see what is happening in the Ykt6 band as it is over-exposed - there could be multiple merged bands in there. A better resolved band or a lower exposure image would be useful to assess what is going on here although ultimately the mass-spec is more useful.
9. It is difficult to assess the robustness of the numeric data throughout as biological and technical replicates are inconsistently described in the figure legends, for example cell numbers vs independent experiment numbers. Quantification of 2C has no information at all. Also what do the bar graphs depict - is it the mean? These bar charts would be better presented as dot plots so that the spread of data and variability can be assessed and this would also help show cell number. This is particularly important e.g. in figure 2B where only 2 independent experiments have been quantified.
10. Figure 5E is lacking any quantification to assess how representative these images are.
11. From the text it is very difficult to determine what the difference is between the non-viable cells expressing Ykt6S182D,S183D in fig 5D and the viable ones in fig 5E.
12. Perhaps a little more discussion to resolve the apparent discrepancies in the role of Ykt6 in autophagosome biogenesis would be useful 'In both ykt6 and dsl3 ts mutants, autophagosomes are complete..... there are much less autophagosomes in both ykt6 and dsl3 ts mutants'. Does this tell us more about what it is doing? Initiation vs maturation.
13. There are a lot of typos, especially in the discussion.

Referee #3:

Gao et al report that Ykt6, a SNARE required for fusion of the autophagosome with the vacuole is recruited at very early stages of phagophore assembly. They found that both the Atg1, Atg13 and Atg17 complex and the Dsl complex is required for recruitment of Ykt6. Using an artificial giant cargo, they show that Ykt6 is, as Atg8 distributed over the whole phagophore. Finally, they test the relevance of a previously identified Atg1-kinase motif within Ykt6.

The story is short, but interesting. It addresses the open question how Ykt6 reaches the autophagosome and is thus interesting to a larger readership. The manuscript is easy to understand and the experiments are both well designed and controlled. The manuscript to my

opinion only requires minor revision prior to publication.

1. The authors propose a model, where Atg1 is both required for recruitment and inhibition of the fusogenic activity of Ykt6. It would be interesting to check an Atg1-kinase dead mutant. Is Ykt6 still recruited, or is the kinase activity also needed for this step?

2. The authors assume that phosphorylation of Ykt6 by Atg1 selectively inhibits its function at autophagosomes. I do not understand why the phosphomimetic Ykt6 mutants are then affecting viability. If these sites are only phosphorylated by Atg1, the other essential functions should stay intact.

3. Most likely this is beyond the scope of this study, but the identification of an autophagy-specific phosphatase regulating Ykt6 would significantly strengthen the manuscript.

We would like to thank reviewers for their insightful comments, which helped us to improve our manuscript. To answer the criticisms, we have

- Provided evidence that Ykt6 is partially required for Cvt pathway (Figure S1).
- Provided evidence that Dsl1 complex does not travel with Ykt6 to the PAS under starvation conditions (Figure 3G).
- Added a wild-type control to Figure 3A and Figure 3G (now Figure 4A)
- Added the quantification of Figure 3G (now Figure 4A) in Figure 4B to show that both the COPII-coated vesicles and the Dsl1 complex contribute to the targeting of Ykt6 to the PAS.
- Provided evidence that at the restrictive temperature, the overall ER morphology and ERES distribution do not change in *tip20 ts* mutant cells, and the accumulated Ykt6 does not localize to either the ER or the COPII vesicles (Figure 4C).
- Demonstrated that the kinase activity of Atg1 is required for the recruitment of Ykt6 to the PAS by using *Atg1^{D211A}* kinase-dead mutant (Figure 6D, E).
- Added the quantification of Figure 5E (now Figure 7B).
- Demonstrated by our *in vitro* autophagosome-vacuole fusion assay that Atg1-Atg13 kinase complex can inhibit fusion of autophagosomes carrying wild-type Ykt6, but not the Ykt6 S-A mutant, which lacks the phosphorylation site targeted by Atg1 (Figure 7D, E).
- Adjusted and expanded the text to clarify further points raised.

Please find below a detailed response to all specific comments of the reviewers.

Referee#1:

In this report, the authors show that GFP-Ykt6 is recruited to the PAS and autophagosomes in an Atg1 and Dsl1 complex-dependent manner. Although Ykt6 is present on the phagophore, the authors propose that its fusogenic activity is kept inhibited by phosphorylation of S182 and S183 by Atg1 until required. Accordingly, overexpression of the Ykt6S182D/S183D mutant blocks autophagosome-vacuole fusion.

Overall, this is solid work, and the data are clear and well-presented.

Thank you for the overall positive evaluation.

Major concerns

1. The authors hypothesize that phosphorylated Ykt6 is non-fusogenic, but the evidence for this is not strong. As the authors have developed a well-working *in vitro* fusion assay system, it would be possible to test the effect of Atg1-dependent phosphorylation of Ykt6 on autophagosomal fusion (by combining the experiments in Fig. 3H and Fig. 5C). Alternatively, if Ykt6^{S182A/S183A} is constitutively active, it would be worth testing whether premature fusion of a phagophore with the vacuole occurs (e.g., by live-cell imaging of Ape1-overexpressing cells).

We thank the reviewer for these important points. As the reviewer suggested, we isolated autophagosomes from Ykt6 wild-type and non-phosphorylatable *ykt6*^{S182A/S183A} mutant cells and performed our *in vitro* autophagosome-vacuole fusion assay in the presence or absence of purified Atg1-Atg13 kinase complex. In agreement with the *in vivo* data, the purified autophagosomes from cells expressing either wild-type or mutant Ykt6 could fuse with the isolated vacuoles. Importantly, the addition of purified Atg1-Atg13 kinase complex together with ATP strongly inhibited fusion of autophagosomes from cells expressing wild-type Ykt6. In contrast, the fusion of autophagosomes from cells expressing the *ykt6*^{S182A/S183A} mutant with vacuoles was resistant to the Atg1-Atg13 kinase complex addition (Figure 7D, E). These results clearly demonstrate that the Atg1-Atg13 kinase complex blocks the fusogenic activity of Ykt6 on autophagosomes by phosphorylating this SNARE.

2. It is not demonstrated whether Ykt6 on the phagophore membrane is indeed phosphorylated. It would be ideal to determine by mass spectrometry whether phosphorylated Ykt6 is enriched in the phagophore or autophagosomal fractions in an Atg1-dependent manner.

We agree with the reviewer that it would be perfect if we could detect the phosphorylated Ykt6 on the phagophore by mass spectrometry. However, it is extremely challenging to test this as only a fraction of Ykt6 is recruited to the phagophore under starvation conditions, and phosphorylation is expected to last only until Atg1 is inactivated (Gao et al., 2018b; Figure 2C). We nevertheless analyzed if

we could detect Ykt6 on purified autophagosomes using a protein mass spectrometry approach, yet failed to see clear enrichment of autophagosome-specific proteins.

However, our *in vitro* assay favors the idea that Ykt6 is indeed phosphorylated as long as the Atg1-Atg13 kinase complex is active, but becomes fusogenic once autophagosomes are mature and Atg1-Atg13 is removed and/or inactivated (Figure 7D, E).

3. The data in Fig. 3G is one of the most important data in this study. Some quantification is required (including a wild-type control). A wild-type control is also missing and should be included in Fig. 3A.

We have added the quantification and the wild-type control as requested.

4. The data in Fig. 2A and B show that Atg17 is required for Ykt6 recruitment. Given that Atg17 is not required for the Cvt pathway, is Ykt6 dispensable for Cvt?

We tested whether Ykt6 is required for Cvt pathway by following Ape1 processing (Figure S1). Our data shows that Ykt6 is partially involved in the Cvt pathway.

Minor points

Why is there less accumulation of autophagosomes in the absence of Ykt6 or the Dsl1 complex (Fig. 4B)? Some explanation is required.

We have added the explanation to the text.

P. 11: (Figure 3F, G) should be (Figure 3H, L). The same for its legend.

P. 16: (Figure 5G) should be (Figure 5F).

We apologize for our mistakes. They have been corrected.

Referee #2:

In this study the authors build on their previous works identifying Ykt6 as an R-SNARE on autophagosomes. They demonstrate that Ykt6 localises to the autophagosome upon nitrogen starvation in yeast, as well as several other organelles, and that this occurs early during the initiation of autophagosome formation in a manner dependent on Atg1 protein kinase complex. Ykt6, as well as the ER-resident Dsl-1 complex are then shown to be required for autophagosome/lysosome fusion. The data for this are clear and well presented. Lead by their previous finding that Ykt6 can interact with Dsl-1, the authors then show that Dsl-1 and COPII are required for Ykt6 delivery to the phagophore. In Dsl-1 mutant cells, autophagosomes form but are fusion incompetent, and in the latter, autophagosome biogenesis is inhibited. Finally, the authors show that premature Ykt6-mediated fusion of the autophagosome with lysosomes/vacuoles is prevented by ATG1-dependent phosphorylation.

The strengths of this paper lie in the characterisation of Ykt6 recruitment to the phagophore and its regulation by phosphorylation to control the timing of fusion with the vacuole. This adds a key detail to current models of the fusion of interest to the autophagy field and those interested in SNARE regulation. On the other hand, whilst Dsl-1 does seem to be required for Ykt6 localisation to the phagosome, the data describing a more specific role for Dsl-1 in targeting Ykt6 are compelling but indirect. The evidence for COPII's role in this process seems even more circumstantial. Thus this section does not progress knowledge much beyond what could already be deduced from the current literature, but if substantially improved would be more generally of interest to the membrane trafficking field. This would require a significant amount of work however which may not be feasible in the current crisis.

Major concerns

The authors propose that 'Our data support a model where Ykt6 interaction with the ER resident Dsl1 complex is a prerequisite for its targeting possibly via COPII coated vesicles to the PAS at the very early stage of autophagosome biogenesis'. The advancement here is that Dsl-1 is required for Ykt6 targeting to the autophagosome, since these interactions are already known. However, the evidence for this seems

largely circumstantial and conclusions poorly supported as the data could be interpreted in a number of ways.

1. Whilst Dsl-1 activity is shown to be required, a requirement for a direct Dsl-1-Ykt6 interaction has not been demonstrated, for example with mutations selectively interrupting this interface.

We agree with the reviewer that such a mutant would be extremely informative, yet we feel that such an analysis is beyond the scope of this study. Finding such an interface along the Dsl1 complex (three subunits) would be challenging and specific interfaces were only recently shown by Travis et al., JBC 2020. In turn, identifying Ykt6 mutants that impair Dsl1 binding might block the function of its SNARE domain and thus fusion. We consider our observations that Dsl mutants have strongly impaired Ykt6 localization to autophagosomes as a novel link that has sufficient value on its own to be reported here.

2. Given the known role of Dsl-1 as an ER-resident membrane tether for ER-ER fusions and retrograde COPI transport, the requirement for Dsl-1 in Ykt6 targeting to the phagophore could easily be due to indirect effects on the structure of, and membrane flux through, the secretory pathway. This could explain why autophagosome abundance is reduced if membrane is less freely available or dynamic. Alternatively, Dsl-1 could be important for the recycling of machinery required to transport Ykt6 to the phagophore but not Ykt6 itself. The authors do not appear to have looked at the health of the secretory pathway in the absence of Dsl-1 or the structure of the ER/ERES, for example with secretion assays or EM of ER and Golgi structures.

We thank the reviewer for these important points. To test the health of the secretory pathway in the Dsl mutant, we determined the ER morphology (Sec63-3xmCherry) and ERES distribution (Sec13-3xmCherry) by fluorescence microscopy in the *tip20 ts* mutant under starvation conditions at either permissive or restrictive temperature cells (Figure 4C). We observed that both ER morphology and ERES distribution remain unperturbed also at restrictive temperature, although the localization of Ykt6

changes completely (Figure 4C). These data indicate that mutants interfering with Dsl1 complex function do not perturb the ER morphology, although they interfere with the trafficking of Ykt6.

3. Ykt6 on ERGIC membranes has been shown to be recruited to ERES by TANGO1 (Santos) - it is therefore possible that Dsl-1 is similarly required for the retrograde transport of Ykt6 back to the ER for re-routing to the phagophore. The authors have not looked to see whether Ykt6 is still localised to ER membranes in the Dsl-1 mutants which would help decipher whether Dsl-1 is required for general ER localisation of Ykt6 or more specifically involved in Ykt6 targeting to COPII vesicles destined for the phagophore.

We thank the reviewer also for these important points. To test whether Ykt6 is still localized to ER membranes in the Dsl1 complex mutants, we colocalized GFP-tagged Ykt6 with Sec63-3xmCherry in the *tip20 ts* mutant under starvation conditions at either permissive or restrictive temperature (Figure 4C). Ykt6 punta only colocalized with Sec63 at permissive temperature, but not at higher temperature. This result supports the notion that the Dsl1 complex is required to confer Ykt6 localization to the ER.

4. Although an interaction between Ykt6 and Dsl-1 has been shown before, the authors do not investigate this directly here. The Ykt6-Dsl-1 interaction appears weaker than that between Dsl-1 and other SNAREs (Meiringer et al) so is it influenced by conditions that induce autophagy - for example is this interaction more prevalent following nitrogen starvation? Is Ykt6 modified on starvation so that the interaction is enhanced? Answers to these questions would demonstrate a more direct link and could provide insight into how Ykt6 is sent to the phagophore instead of the Golgi when autophagy is induced.

We thank the reviewer for pointing this out. To test whether the interaction between Ykt6 and Dsl-1 is influenced by autophagy-inducing conditions, we tagged Dsl3 with GFP and isolated the Dsl3-GFP fusion from yeast cells after stable isotope labeling by amino acids (SILAC; Ong et al., 2002) under nutrient-rich and starvation conditions. Following affinity purification, eluted proteins were identified by mass

spectrometry. Importantly, and in agreement with our previous data, Ykt6 was co-eluted with the entire Dsl1 complex in both conditions. However, we did not observe any strong change in the interaction of Ykt6 and Dsl3 or any other interactors under starvation conditions, and thus cannot assert about a possible role of the Dsl complex in targeting Ykt6. We speculate that the Dsl complex traps a fraction of Ykt6 at the ER that gets localized to the phagophore.

MS analysis of GFP-tagged Dsl3 and untagged control cells. Intensities are plotted against normalized heavy/light SILAC ratios. Significant outliers are colored in red ($p < 1e-11$), orange ($p < 0.0001$), or blue ($p < 0.05$); other identified proteins are shown in light blue.

5. Autophagosomes from Dsl-1 mutants are fusion incompetent in in vitro assays, as are those from Ykt6 mutants, and thus the authors conclude Dsl-1 targets Ykt6 to autophagosomes for fusion. Whilst this is a logical explanation it is merely inferred from indirect evidence - a rescue by adding Ykt6 to the membranes from Dsl-1 mutants in this assay would demonstrate that it is Ykt6 targeting and not the targeting of some other machinery that is important. As a tether for retrograde ER trafficking, Dsl-1 could be affecting any number of things.

The authors also state 'Ykt6 is not present on the PAS in the mutant impaired in COPII- and COPI-coated vesicle formation (Figure 3). These observations agree with our working model that COPII-coated vesicles carry Ykt6 to the nascent autophagosome'.

The reviewer suggests a very challenging experiment, the addition of purified Ykt6 to isolated autophagosomes of the Dsl3 mutant to restore the fusion with vacuoles. Such an experiment requires that we establish the purification of a prenylated and functional Ykt6, add it to a fusion assay and demonstrate its activity. For the fusion promoting complex between the Rab7-like Ypt7 and its chaperone GDI, it took us five years to get it functional in our assays (Langemeyer et al., 2018; 2020). As much as we would like to do such an assay, we first need to establish conditions to get Ykt6 as a functional protein – a study on its own.

Nevertheless, we show now that the ER morphology and ERES distribution are not affected in the *tip20 ts* mutant under starvation conditions at restrictive temperature (Figure 4C), which agrees with our interpretation that Dsl1 mutants specifically interfere with Ykt6 trafficking to the phagophore.

6. This could again easily be due to general disruption to the secretory pathway. Indeed autophagosome biogenesis is disrupted in the COPII mutants implying much bigger defects that would indirectly affect the recruitment of machinery. Also Ykt6 could be coming from other organelles. So whilst this is a good working model caution should be applied in linking COPII to Ykt6 delivery without further evidence for example here 'In conclusion, in this study we show that the Dsl1 complex and COPII-coated vesicles are determinants for the localization of the SNARE Ykt6 to the autophagosomal intermediates' Demonstration of colocalisation between Ykt6 and COPII components, especially upon nitrogen starvation would help better support this model.

To test whether Ykt6 is loaded onto COPII vesicles, we colocalized GFP-tagged Ykt6 with 3xmCherry tagged Sec13 (a subunit of the COPII-coat) in the *tip20 ts* mutant under starvation conditions at either permissive or restrictive temperature (Figure 4C). Ykt6 colocalized with Sec13 at the permissive temperature, but not at the restrictive temperature (Figure 4C). These data indicate that Ykt6 is loaded onto COPII vesicles, which requires Dsl1 complex.

7. In Figure 3E Dsl-1 and Ykt6 co-localise to a single puncta upon serum starvation -

is this the phagophore? This would imply Dsl-1 travels with Ykt6. Alternatively this could be ER. This should be investigated with double or triple fluorescence labels to determine where they coalesce as it could influence interpretation of models.

We indeed observed some colocalization between Dsl1 and Ykt6 (Figure 3E,F) under starvation conditions. To test whether the Dsl1 complex travel with Ykt6 to the PAS, we colocalized Dsl3-GFP and mCherry-Atg8 under nutrient-rich and starvation conditions. In both cases, we did not observe any colocalization between Dsl3 and Atg8 (Figure 3G), suggesting that the Dsl complex confers Ykt6 localization to the ER, but does not travel to the PAS.

Minor concerns

8. In figure 5C it is difficult to really see what is happening in the Ykt6 band as it is over-exposed - there could be multiple merged bands in there. A better resolved band or a lower exposure image would be useful to assess what is going on here although ultimately the mass-spec is more useful.

We apologize that the figure legend was not clear enough. In fact, we cut the Ykt6 band from the gel and analyzed it by mass-spec to identify the phosphorylation sites of Ykt6. The identified phospho-sites were also found by others. We now added more details to the figure legend.

9. It is difficult to assess the robustness of the numeric data throughout as biological and technical replicates are inconsistently described in the figure legends, for example cell numbers vs independent experiment numbers. Quantification of 2C has no information at all. Also what do the bar graphs depict - is it the mean? These bar charts would be better presented as dot plots so that the spread of data and variability can be assessed and this would also help show cell number. This is particularly important e.g. in figure 2B where only 2 independent experiments have been quantified.

We apologize for any mistake in Figure 2B. Actually, three independent experiments have been quantified. We added more details to relevant figure legends to clarify this.

10. Figure 5E is lacking any quantification to assess how representative these images are.

We have added the quantification now.

11. From the text it is very difficult to determine what the difference is between the non-viable cells expressing Ykt6S182D,S183D in fig 5D and the viable ones in fig 5E.

We have updated the text accordingly.

12. Perhaps a little more discussion to resolve the apparent discrepancies in the role of Ykt6 in autophagosome biogenesis would be useful 'In both ykt6 and dsl3 ts mutants, autophagosomes are complete..... there are much less autophagosomes in both ykt6 and dsl3 ts mutants'. Does this tell us more about what it is doing? Initiation vs maturation.

We agree and have revised the text accordingly.

13. There are a lot of typos, especially in the discussion.

We apologize for any mistakes. We have gone again through the text and checked several times to correct any remaining mistakes.

Referee #3:

Gao et al report that Ykt6, a SNARE required for fusion of the autophagosome with the vacuole is recruited at very early stages of phagophore assembly. They found that both the Atg1, Atg13 and Atg17 complex and the Dsl complex is required for recruitment of Ykt6. Using an artificial giant cargo, they show that Ykt6 is, as Atg8 distributed over the whole phagophore. Finally, they test the relevance of a previously identified Atg1-kinase motif within Ykt6.

The story is short, but interesting. It addresses the open question how Ykt6 reaches the autophagosome and is thus interesting to a larger readership. The manuscript is easy to understand and the experiments are both well designed and controlled. The manuscript to my opinion only requires minor revision prior to publication.

Thank you for the overall positive evaluation.

1. The authors propose a model, where Atg1 is both required for recruitment and inhibition of the fusogenic activity of Ykt6. It would be interesting to check an Atg1-kinase dead mutant. Is Ykt6 still recruited, or is the kinase activity also needed for this step?

We thank the reviewer for this important point. To test whether the recruitment of Ykt6 depends on Atg1 kinase, we colocalized GFP-tagged Ykt6 with mCherry-tagged Atg8 in the wild-type and cells expressing the kinase dead *atg1^{D211A}* under normal and starvation conditions. We observed that in contrast to the wild-type, Ykt6 does not colocalize with Atg8 in *atg1^{D211A}* mutant cells under starvation conditions (Figure 6D, E). This result shows that the kinase activity of Atg1 is also required for Ykt6 targeting to the PAS.

2. The authors assume that phosphorylation of Ykt6 by Atg1 selectively inhibits its function at autophagosomes. I do not understand why the phosphomimetic Ykt6 mutants are then affecting viability. If these sites are only phosphorylated by Atg1, the other essential functions should stay intact.

Ykt6 is required for several trafficking pathways and the phosphorylation sites are located within the SNARE domain. We speculate that phosphomimetic Ykt6 mutants might directly alter the function of the SNARE domain, which will affect all the fusion function of Ykt6 and consequently cell viability.

In support of the direct role of phosphorylation in the control of Ykt6 fusion activity, we now show that autophagosome-vacuole fusion is strongly blocked by addition of the purified Atg1-Atg13 kinase complex. However, a Ykt6 mutant lacking the phosphorylation sites for Atg1 is resistant to the inhibition by the Atg1-Atg13 kinase

complex. This strongly suggests that Ykt6 is the only target, which is inhibited by added Atg1-Atg13 on purified autophagosomes.

3. Most likely this is beyond the scope of this study, but the identification of an autophagy-specific phosphatase regulating Ykt6 would significantly strengthen the manuscript.

This is an excellent point, but we feel that this analysis would be beyond the scope of this study. We are certainly keen to analyze the phosphatase function in future assays. We nevertheless hope that the reviewer appreciates our efforts to strengthen the study as it stands now.

References:

Langemeyer, L., Borchers, A.-C., Herrmann, E., Füllbrunn, N., Han, Y., Perz, A., Auffarth, K., Kümmel, D., and Ungermann, C. (2020). A conserved and regulated mechanism drives endosomal Rab transition. *eLife* 9, 191.

Langemeyer, L., Perz, A., Kümmel, D., and Ungermann, C. (2018). A guanine nucleotide exchange factor (GEF) limits Rab GTPase-driven membrane fusion. *J Biol Chem* 293, 731–739.

Ong, S-E, Blagojev, B, Kratchmarova, I, Kristensen, DB, Steen, H, Pandey, A, and Mann, M (2002). Stable isotope labeling by amino acids in cell culture, SILAC, as a simple and accurate approach to expression proteomics. *Mol Cell Proteomics* 1, 376–386.

Dear Dr. Ungermann

Thank you for the submission of your revised manuscript to EMBO reports. We have now received the full set of referee reports that is copied below.

As you will see, all referees are very positive about the study and suggest only minor textual changes to clarify the role of Dsl-1.

Before I can accept the manuscript, I need you to address some minor points below:

- Please reduce the number of keywords to five. I suggest to remove 'membrane fusion' and 'autophagosome' (or 'autophagy')
- Please add a heading to the Conflict of interest paragraph
- Author contributions: please use abbreviations also for the surnames and specify the contribution of J. Rose
- Reference format: If there are more than 10 authors please list the first 10 authors followed by 'et al'
- Movies: please provide their legends as a separate README.txt files and then zip it together with the movie file.
- Please provide Table EV1 and EV2 as separate Expanded View files.
- Please ensure that all grant numbers listed in the Acknowledgement section are also listed in the online submission system.
- Finally, EMBO reports papers are accompanied online by A) a short (1-2 sentences) summary of the findings and their significance, B) 2-3 bullet points highlighting key results and C) a synopsis image that is 550x200-600 pixels large (width x height) in .png format. You can either show a model or key data in the synopsis image. Please note that the size is rather small and that text needs to be readable at the final size. Please send us this information along with the revised manuscript.

With kind regards,

Martina

Martina Rembold, PhD
Editor
EMBO reports

Referee #1:

The authors have responded appropriately to the previous concerns. The new in vitro data in Fig. 7 significantly strengthened the authors' model.

Referee #2:

The authors have adequately addressed the points I raised on initial review, and those of the other reviewers.

I have just one point to raise which the authors may choose to consider or ignore, and that is regarding the clarity of their interpretations of the new data surrounding ER localisation of Ykt6. The authors write: 'we colocalized GFP-tagged Ykt6 with 3xmCherry-tagged Sec63 (a marker protein of the ER) and Sec13 (a subunit of the COPII coat) in tip20 ts mutant at either permissive or restrictive temperature.... At permissive temperature, Ykt6 puncta colocalized with Sec63 and Sec13, but this colocalization was lost at the restrictive temperature (Figure 4C). Taken together, these data suggest that mutations in subunits of the Dsl1 complex specifically block efficient delivery of Ykt6, possibly through COPII vesicles, to the phagophore'.

This is not quite what these specific data show. A clearer interpretation of this data is that Dsl-1 is required for the recruitment of Ykt6 to the ER/COPII vesicles.

I acknowledge this interpretation has been made in the discussion but consistency through the text may help the reader:

'We speculate that the Dsl1 complex, which does not travel with Ykt6 to the PAS (Figure 3G), maintains a pool of Ykt6 at the ER membrane and makes it available for loading onto COPII-coated vesicles (Figure 7F)'

Referee #3:

The authors have answered all my questions.

The authors have addressed all minor editorial requests.

Dr. Christian Ungermann
University of Osnabrück
Biology/Chemistry
Barbarastrasse 13
Osnabrück 49076
Germany

Dear Christian,

Thank you for submitting your revised manuscript. I have now looked at everything and am pleased to accept it for publication in the next available issue of EMBO reports.

Congratulations on a nice study!

At the end of this email I include important information about how to proceed. Please ensure that you take the time to read the information and complete and return the necessary forms to allow us to publish your manuscript as quickly as possible.

As part of the EMBO publication's Transparent Editorial Process, EMBO reports publishes online a Review Process File to accompany accepted manuscripts. As you are aware, this File will be published in conjunction with your paper and will include the referee reports, your point-by-point response and all pertinent correspondence relating to the manuscript.

If you do NOT want this File to be published, please inform the editorial office within 2 days, if you have not done so already, otherwise the File will be published by default [contact: emboreports@embo.org]. If you do opt out, the Review Process File link will point to the following statement: "No Review Process File is available with this article, as the authors have chosen not to make the review process public in this case."

Should you be planning a Press Release on your article, please get in contact with emboreports@wiley.com as early as possible, in order to coordinate publication and release dates.

Please note that under the DEAL agreement of German scientific institutions with our publisher Wiley, your paper might be eligible for open access publication in a way that is free of charge for the authors. Please contact either the administration at your institution or our publishers at Wiley (emboreports@wiley.com) for further questions

<https://authorservices.wiley.com/author-resources/Journal-Authors/open-access/affiliation-policies-payments/institutional-funder-payments.html>

Thank you again for your contribution to EMBO reports and congratulations on a successful publication. Please consider us again in the future for your most exciting work.

Kind regards,
Martina

THINGS TO DO NOW:

You will receive proofs by e-mail approximately 2-3 weeks after all relevant files have been sent to our Production Office; you should return your corrections within 2 days of receiving the proofs.

Please inform us if there is likely to be any difficulty in reaching you at the above address at that time. Failure to meet our deadlines may result in a delay of publication, or publication without your corrections.

All further communications concerning your paper should quote reference number EMBOR-2020-50733V3 and be addressed to emboreports@wiley.com.

Should you be planning a Press Release on your article, please get in contact with emboreports@wiley.com as early as possible, in order to coordinate publication and release dates.

Corresponding Author Name: Christian Ungermann

Manuscript Number: EMBOR-2020-50733V1